# First-order Stochastic Algorithms for Escaping From Saddle Points in Almost Linear Time

**Yi Xu[†], Rong Jin[‡], Tianbao Yang[†]**
[†] Department of Computer Science, The University of Iowa, Iowa City, IA 52246, USA
[‡] Machine Intelligence Technology, Alibaba Group, Bellevue, WA 98004, USA
`{yi-xu, tianbao-yang}@uiowa.edu, jinrong.jr@alibaba-inc.com`

## Abstract

In this paper, we consider first-order methods for solving stochastic non-convex optimization problems. The key building block of the proposed algorithms is first-order procedures to extract negative curvature from the Hessian matrix through a principled sequence starting from noise, which are referred to *NEgative-curvature-Originated-from-Noise or NEON* and are of independent interest. Based on this building block, we design purely first-order stochastic algorithms for escaping from non-degenerate saddle points with a much better time complexity (almost linear time in the problem's dimensionality) under a bounded variance condition of stochastic gradients than previous first-order stochastic algorithms. In particular, we develop a general framework of *first-order stochastic algorithms* with a second-order convergence guarantee based on our new technique and existing algorithms that may only converge to a first-order stationary point. For finding a nearly *second-order stationary point* $\mathbf{x}$ such that $\|\nabla F(\mathbf{x})\| \leq \epsilon$ and $\nabla^2 F(\mathbf{x}) \geq -\sqrt{\epsilon}I$ (in high probability), the best time complexity of the presented algorithms is $\widetilde{O}(d/\epsilon^{3.5})$, where $F(\cdot)$ denotes the objective function and $d$ is the dimensionality of the problem. To the best of our knowledge, this is the first theoretical result of first-order stochastic algorithms with an almost linear time in terms of problem's dimensionality for finding second-order stationary points, which is even competitive with existing stochastic algorithms hinging on the second-order information.

## 1 Introduction

The problem of interest in this paper is Stochastic Non-Convex Optimization given by

$$\min_{\mathbf{x}\in\mathbb{R}^d} F(\mathbf{x}) \triangleq \mathrm{E}_\xi[f(\mathbf{x};\xi)], \tag{1}$$

where $\xi$ is a random variable and $f(\mathbf{x};\xi)$ is a random smooth non-convex function of $\mathbf{x}$. The only information available of $F(\mathbf{x})$ to us is sampled stochastic functions $f(\mathbf{x};\xi)$ and their gradients.

A popular choice of algorithms for solving (1) is (mini-batch) stochastic gradient descent (SGD) method and its variants [6]. However, these algorithms do not necessarily guarantee to escape from a saddle point (more precisely a non-degenerate saddle point) $\mathbf{x}$ satisfying that: $\nabla F(\mathbf{x}) = 0$ and the minimum eigen-value of $\nabla^2 F(\mathbf{x})$) is less than 0. Recently, new variants of SGD by adding isotropic noise into the stochastic gradient were proposed (noisy SGD [5], stochastic gradient Langevin dynamics (SGLD) [23]). These two works provide rigorous analyses of the noise-injected update for escaping from a saddle point. Unfortunately, both variants suffer from a polynomial time complexity with a super-linear dependence on the dimensionality $d$ (at least a power of 4), which renders them not practical for optimizing problems of high dimension.

On the other hand, second-order information carried by the Hessian has been utilized to escape from a saddle point, which usually yields an almost linear time complexity in terms of the dimensionality $d$ under the assumption that the Hessian-vector product (HVP) can be performed in a linear time. In

Table 1: Comparison with existing **Stochastic Algorithms** for achieving an $(\epsilon, \gamma)$-SSP to (1), where $p$ is a number at least $4$, IFO (incremental first-order oracle) and ISO (incremental second-order oracle) are terminologies borrowed from [20], representing $\nabla f(\mathbf{x}; \xi)$ and $\nabla^2 f(\mathbf{x}; \xi)\mathbf{v}$ respectively, $T_h$ denotes the runtime of ISO and $T_g$ denotes the runtime of IFO. $\widetilde{O}(\cdot)$ hides a poly-logarithmic factor. SM refers to stochastic momentum methods. For $\gamma$, we only consider as lower as $\epsilon^{1/2}$.

| Algorithm | Oracle | Target | Time Complexity |
|---|---|---|---|
| Noisy SGD [5] | IFO | $(\epsilon, \epsilon^{1/2})$-SSP, high probability | $\widetilde{O}\left(T_g d^p \epsilon^{-p}\right)$ |
| SGLD [23] | IFO | $(\epsilon, \epsilon^{1/2})$-SSP, high probability | $\widetilde{O}\left(T_g d^p \epsilon^{-4}\right)$ |
| Natasha2 [1] | IFO + ISO | $(\epsilon, \epsilon^{1/2})$-SSP, expectation | $\widetilde{O}\left(T_g \epsilon^{-3.5} + T_h \epsilon^{-2.5}\right)$ |
| Natasha2 [1] | IFO + ISO | $(\epsilon, \epsilon^{1/4})$-SSP, expectation | $\widetilde{O}\left(T_g \epsilon^{-3.25} + T_h \epsilon^{-1.75}\right)$ |
| SNCG [17] | IFO + ISO | $(\epsilon, \epsilon^{1/2})$-SSP, high probability | $\widetilde{O}\left(T_g \epsilon^{-4} + T_h \epsilon^{-2.5}\right)$ |
| SVRG-Hessian [20] (finite-sum objectives) ($n$ is number of components) | IFO + ISO | $(\epsilon, \epsilon^{1/2})$-SSP, high probability | $\widetilde{O}\left(T_g (n^{2/3}\epsilon^{-2} + n\epsilon^{-1.5}) + T_h(n\epsilon^{-1.5} + n^{3/4}\epsilon^{-7/4})\right)$ |
| NEON-SGD, NEON-SM (**this work**) | IFO | $(\epsilon, \epsilon^{1/2})$-SSP, high probability | $\widetilde{O}\left(T_g \epsilon^{-4}\right)$ |
| NEON-SCSG (**this work**) | IFO | $(\epsilon, \epsilon^{1/2})$-SSP, high probability | $\widetilde{O}\left(T_g \epsilon^{-3.5}\right)$ |
| NEON-SCSG (**this work**) | IFO | $(\epsilon, \epsilon^{4/9})$-SSP, high probability | $\widetilde{O}\left(T_g \epsilon^{-3.33}\right)$ |
| NEON-Natasha (**this work**) | IFO | $(\epsilon, \epsilon^{1/2})$-SSP, expectation | $\widetilde{O}\left(T_g \epsilon^{-3.5}\right)$ |
| NEON-Natasha (**this work**) | IFO | $(\epsilon, \epsilon^{1/4})$-SSP, expectation | $\widetilde{O}\left(T_g \epsilon^{-3.25}\right)$ |
| NEON-SVRG (**this work**) (finite sum) | IFO | $(\epsilon, \epsilon^{1/2})$-SSP, high probability | $\widetilde{O}\left(T_g \left(n^{2/3}\epsilon^{-2} + n\epsilon^{-1.5} + \epsilon^{-2.75}\right)\right)$ |

practice, HVP can be estimated by a finite difference approximation using two gradient evaluations. However, the rigorous analysis of algorithms using such noisy approximation for solving non-convex optimization remains unsolved, and heuristic approaches may suffer from numerical issues. Although for some problems with special structures (e.g., neural networks), HVP can be efficiently computed using gradients, a HVP-free method that can escape saddle points for a broader family of non-convex problems is still desirable.

This paper aims to design HVP-free stochastic algorithms for solving (1), which can converge to second order stationary points with a time complexity that is almost linear in the problem's dimensionality. Our main contributions are:

- As a key building block of proposed algorithms, first-order procedures (NEON) are proposed to extract negative curvature from the Hessian using a principled sequence starting from noise. Interestingly, our perspective of NEON connects the existing two classes of methods (noise-based and HVP-based) for escaping from saddle points. We provide a formal analysis of simple procedures based on gradient descent and accelerated gradient method for exacting a negative curvature direction from the Hessian.

- We develop a general framework of first-order algorithms for stochastic non-convex optimization by combining the proposed first-order NEON procedures to extract negative curvature with existing first-order stochastic algorithms that aim at a first-order critical point. We also establish the time complexities of several interesting instances of our general framework for finding a nearly $(\epsilon, \gamma)$-second-order stationary point (SSP), i.e., $\|\nabla F(\mathbf{x})\| \leq \epsilon$, and $\lambda_{\min}(\nabla^2 F(\mathbf{x})) \geq -\gamma$, where $\|\cdot\|$ represents Euclidean norm of a vector and $\lambda_{\min}(\cdot)$ denotes the minimum eigen-value. A summary of our results and existing results for Stochastic Non-Convex Optimization is presented in Table 1.

## 2 Other Related Work

SGD and its many variants (e.g., mini-batch SGD and stochastic momentum (SM) methods) have been analyzed for stochastic non-convex optimization [6, 7, 8, 22]. The iteration complexities of all these algorithms is $O(1/\epsilon^4)$ for finding a first-order stationary point (FSP) (in expectation $\mathrm{E}[\|\nabla F(\mathbf{x})\|_2^2] \leq \epsilon^2$ or in high probability). Recently, there are some improvements for stochastic non-convex optimization. [14] proposed a first-order stochastic algorithm (named SCSG) using the variance-reduction technique, which enjoys an iteration complexity of $O(1/\epsilon^{-10/3})$ for finding an FSP (in expectation), i.e., $\mathrm{E}[\|\nabla F(\mathbf{x})\|_2^2] \leq \epsilon^2$. [1] proposed a variant of SCSG (named Natasha1.5) with the same convergence and complexity. An important application of NEON is that previous stochastic algorithms that have a first-order convergence guarantee can be strengthened to enjoy a second-order convergence guarantee by leveraging the proposed first-order NEON procedures to escape from saddle points. We will analyze several algorithms by combining the updates of SGD, SM, and SCSG with the proposed NEON.

Several recent works [17, 1, 20] propose to strengthen existing first-order stochastic algorithms to have second-order convergence guarantee by leveraging the second-order information. [17] used mini-batch SGD, [20] used SVRG for a finite-sum problem, and [1] used a similar algorithm to SCSG for their first-order algorithms. The second-order methods used in these studies for computing negative curvature can be replaced by the proposed NEON procedures. It is notable although a generic approach for stochastic non-convex optimization was proposed in [20], its requirement on the first-order stochastic algorithms precludes many interesting algorithms such as SGD, SM, and SCSG. Stronger convergence guarantee (e.g., converging to a global minimum) of stochastic algorithms has been studied in [9] for a certain family of problems, which is beyond the setting of the present work.

It is also worth mentioning that the field of non-convex optimization is moving so fast that similar results have appeared online after the preliminary version of this work [2]. Allen-Zhu and Li [2] proposed NEON2 for finding a negative curvature, which includes a stochastic version and a deterministic version. We notice several differences between the two works: (i) they used Gaussian random noise with a variance proportional to $d^{-C}$, where $C$ is a large unknown constant, in contrast our NEON and NEON+ procedures use random noise sampled from the sphere of an Euclidean ball with radius proportional to $\log^{-2}(d)$; (ii) the update of their deterministic NEON2$^{\text{det}}$ is constructed based on the Chebyshev polynomial, in contrast our NEON+ with a similar iteration complexity is based on the well-known Nesterov's accelerated gradient method; (iii) we provide a general framework/analysis for promoting first-order algorithms to enjoy second-order convergence, which could be useful for promoting new first-order stochastic algorithms; (iv) the reported iteration complexity of their NEON2$^{\text{online}}$ is better than our stochastic variants of NEON. However, in most cases the total complexity for finding an $(\epsilon, \sqrt{\epsilon})$-SSP is dominated by the complexity for finding a stationary point not by the complexity of stochastic NEON for finding a negative curvature.

## 3 Preliminaries

Let $\|\cdot\|$ denote the Euclidean norm of a vector and $\|\cdot\|_2$ denote the spectral norm of a matrix. Let $\mathbf{S}_r^d$ denote the sphere of an Euclidean ball centered at zero with radius $r$, and $[t]$ denote a set $\{0, \ldots, t\}$. A function $f(\mathbf{x})$ has a $L_1$-Lipschitz continuous gradient if it is differentiable and there exists $L_1 > 0$ such that $\|\nabla f(\mathbf{x}) - \nabla f(\mathbf{y})\| \leq L_1 \|\mathbf{x} - \mathbf{y}\|$ holds for any $\mathbf{x}, \mathbf{y} \in \mathbb{R}^d$. A function $f(\mathbf{x})$ has a $L_2$-Lipschitz continuous Hessian if it is twice differentiable and there exists $L_2 > 0$ such that $\|\nabla^2 f(\mathbf{x}) - \nabla^2 f(\mathbf{y})\|_2 \leq L_2 \|\mathbf{x} - \mathbf{y}\|$ holds for any $\mathbf{x}, \mathbf{y} \in \mathbb{R}^d$. It implies that $|f(\mathbf{x}) - f(\mathbf{y}) - \nabla f(\mathbf{y})^\top (\mathbf{x} - \mathbf{y}) - \frac{1}{2}(\mathbf{x} - \mathbf{y})^\top \nabla^2 f(\mathbf{y})(\mathbf{x} - \mathbf{y})| \leq \frac{L_2}{6} \|\mathbf{x} - \mathbf{y}\|^3$, and

$$\|\nabla f(\mathbf{x} + \mathbf{u}) - \nabla f(\mathbf{x}) - \nabla^2 f(\mathbf{x})\mathbf{u}\| \leq L_2 \|\mathbf{u}\|^2 / 2. \qquad (2)$$

We first make the following assumptions regarding the problem (1).

**Assumption 1.** *For the problem (1), we assume that*

*(i). every random function $f(\mathbf{x}; \xi)$ is twice differentiable, and it has $L_1$-Lipschitz continuous gradient and $L_2$-Lipschitz continuous Hessian.*

*(ii). given an initial point $\mathbf{x}_0$, there exists $\Delta < \infty$ such that $F(\mathbf{x}_0) - F(\mathbf{x}_*) \leq \Delta$, where $\mathbf{x}_*$ denotes the global minimum of (1).*

*(iii). there exists $G > 0$ such that $\mathbb{E}[\exp(\|\nabla f(\mathbf{x}; \xi) - \nabla F(\mathbf{x})\|^2 / G^2)] \leq \exp(1)$ holds.*

**Remark.** (1) the analysis of NEON or NEON$^+$ or their stochastic versions for extracting the negative curvature only requires Assumption 1 (i). Indeed, the Lipschitz continuous Hessian can be relaxed to locally Lipchitz continuous Hessian condition according to our analysis. (2) Assumptions 1 (ii) (iii) are used in the analysis of Section 5, which are standard assumptions made in the literature of stochastic non-convex optimization [6, 7, 8]. Assumption 1 (iii) implies that $\mathbb{E}[\|\nabla f(\mathbf{x}; \xi) - \nabla F(\mathbf{x})\|^2] \leq V \triangleq G^2$ holds. For stating our time complexities, we assume $G$ is independent of $d$ for finding an approximate local minimum in Section 5. Nevertheless, our comparison of the proposed algorithms with previous algorithms (e.g., SGLD [23], SNCG [17], Natasha2 [1]) in the stochastic setting are fair because similar assumptions are also made. We also note that [5] makes a stronger assumption about the stochastic gradients, i.e., $\|\nabla f(\mathbf{x}; \xi) - \nabla F(\mathbf{x})\| \leq O(d)$, which leads to a worse dependence of time complexity on $d$, i.e., $O(d^p)$ with $p \geq 4$.

Next, we discuss a second-order method based on HVPs to escape from a non-degenerate saddle point $\mathbf{x}$ of a function $f(\mathbf{x})$ that satisfies $\lambda_{\min}(\nabla^2 f(\mathbf{x})) \leq -\gamma$, which can be found in many previous studies [21, 16, 4]. The method is based on a negative curvature (**NC for short is used in the sequel**)

direction $\mathbf{v} \in \mathbb{R}^d$ that satisfies $\|\mathbf{v}\| = 1$ and

$$\mathbf{v}^\top \nabla^2 f(\mathbf{x}) \mathbf{v} \leq -c\gamma, \tag{3}$$

where $c > 0$ is a constant. Given such a vector $\mathbf{v}$, we can update the solution according to

$$\mathbf{x}_+ = \mathbf{x} - \frac{c\gamma}{L_2} \operatorname{sign}(\mathbf{v}^\top \nabla f(\mathbf{x})) \mathbf{v}, \text{ or } \mathbf{x}_+' = \mathbf{x} - \frac{c\gamma}{L_2} \bar{\xi} \mathbf{v}, \tag{4}$$

where $\bar{\xi} \in \{1, -1\}$ is a Rademacher random variable used when $\nabla f(\mathbf{x})$ is not available. The following lemma establishes that the objective value of $\mathbf{x}_+$ or $\mathbf{x}_+'$ is less than that of $\mathbf{x}$ by a sufficient amount, which makes it possible to escape from the saddle point $\mathbf{x}$.

**Lemma 1.** *For $\mathbf{x}$ satisfying $\lambda_{\min}(\nabla^2 f(\mathbf{x})) \leq -\gamma$ and $\mathbf{v}$ satisfying (3), let $\mathbf{x}_+$, $\mathbf{x}_+'$ be given in (4), then we have $f(\mathbf{x}) - f(\mathbf{x}_+) \geq \frac{c^3\gamma^3}{3L_2^2}$ and $\mathrm{E}[f(\mathbf{x}) - f(\mathbf{x}_+')] \geq \frac{c^3\gamma^3}{3L_2^2}$.*

To compute a NC direction $\mathbf{v}$ that satisfies (3), we can employ the Lanczos method or the Power method for computing the maximum eigen-vector of the matrix $(I - \eta\nabla^2 f(\mathbf{x}))$, where $\eta L_1 \leq 1$ such that $I - \eta\nabla^2 f(\mathbf{x}) \succeq 0$. The Power method starts with a random vector $\mathbf{v}_1 \in \mathbb{R}^d$ (e.g., drawn from a uniform distribution over the unit sphere) and iteratively compute $\mathbf{v}_{\tau+1} = (I - \eta\nabla^2 f(\mathbf{x})) \mathbf{v}_\tau, \tau = 1, \ldots, t$. Following the results in [13], it can be shown that if $\lambda_{\min}(\nabla^2 f(\mathbf{x})) \leq -\gamma$, then with at most $\frac{\log(d/\delta^2) L_1}{\gamma}$ HVPs, the Power method finds a vector $\hat{\mathbf{v}}_t = \mathbf{v}_t / \|\mathbf{v}_t\|$ such that $\hat{\mathbf{v}}_t^\top \nabla^2 f(\mathbf{x}) \hat{\mathbf{v}}_t \leq -\frac{\gamma}{2}$ holds with high probability $1 - \delta$. Similarly, the Lanczos method (e.g., Lemma 11 in [21]) can find such a vector $\hat{\mathbf{v}}_t$ with a lower number of HVPs, i.e., $\min(d, \frac{\log(d/\delta^2)\sqrt{L_1}}{2\sqrt{2\varepsilon}})$.

## 4 Key Building Block: Extracting NC From Noise

Our HVP-free stochastic algorithms with provable guarantees for solving (1) presented in next section are based on a key building block, i.e., extracting NC from noise using only first-order information. To tackle the stochastic objective in (1), our method is to compute a NC based on a mini-batch of functions $\sum_{i=1}^m f(\mathbf{x}; \xi_i)/m$ for a sufficiently large number of samples. Thus, a key building block of the proposed method is a first-order procedure to extract NC for a non-convex function $f(\mathbf{x})$ [1].

Below, we first propose a gradient descent based method for extracting NC, which achieves a similar *iteration complexity* to the Power method. Second, we present an accelerated gradient method to extract the NC to match the iteration complexity of the Lanczos method. Finally, we discuss the application of these procedures for stochastic non-convex optimization using mini-batch.

### 4.1 Extracting NC by NEON

The NEON is inspired by the perturbed gradient descent (PGD) method (a method for solving deterministic non-convex problems) proposed in the seminal work [11] and its connection with the Power method as discussed shortly. Around a saddle point $\mathbf{x}$, the PGD method first generates a random noise vector $\hat{\mathbf{e}}$ from the sphere of an Euclidean ball with a proper radius, then starts with a noise perturbed solution $\mathbf{x}_0 = \mathbf{x} + \hat{\mathbf{e}}$, the PGD generates the following sequence of solutions:

$$\mathbf{x}_\tau = \mathbf{x}_{\tau-1} - \eta\nabla f(\mathbf{x}_{\tau-1}). \tag{5}$$

To establish a connection with the Power method and motivate the proposed NEON, let us define another sequence of $\hat{\mathbf{x}}_\tau = \mathbf{x}_\tau - \mathbf{x}$. Then we have the recurrence for $\hat{\mathbf{x}}_\tau = \hat{\mathbf{x}}_{\tau-1} - \eta\nabla f(\hat{\mathbf{x}}_{\tau-1} + \mathbf{x})$, $\tau = 1, \ldots, t$. It is clear that for $\tau = 1, \ldots, t$,

$$\hat{\mathbf{x}}_\tau = \hat{\mathbf{x}}_{\tau-1} - \eta\nabla f(\mathbf{x}) - \eta(\nabla f(\hat{\mathbf{x}}_{\tau-1} + \mathbf{x}) - \nabla f(\mathbf{x})).$$

To understand the above update, we adopt the following approximation: $\nabla f(\mathbf{x}) \approx 0$ for an approximate saddle point, and from the Lipschitz continuous Hessian condition (2), we can see that $\nabla f(\hat{\mathbf{x}}_{\tau-1} + \mathbf{x}) - \nabla f(\mathbf{x}) \approx \nabla^2 f(\mathbf{x})\hat{\mathbf{x}}_{\tau-1}$ as long as $\|\hat{\mathbf{x}}_{\tau-1}\|$ is small. Then for $\tau = 1, \ldots, t$,

$$\hat{\mathbf{x}}_\tau \approx \hat{\mathbf{x}}_{\tau-1} - \eta\nabla^2 f(\mathbf{x})\hat{\mathbf{x}}_{\tau-1} = (I - \eta\nabla^2 f(\mathbf{x}))\hat{\mathbf{x}}_{\tau-1}.$$

It is obvious that the above approximated recurrence is close to the the sequence generated by the Power method with the same starting random vector $\hat{\mathbf{e}} = \mathbf{v}_1$. This intuitively explains that why the updated solution $\mathbf{x}_t = \mathbf{x} + \hat{\mathbf{x}}_t$ can decrease the objective value due to that $\hat{\mathbf{x}}_t$ is close to a NC of the

**Algorithm 1** NEON($f, \mathbf{x}, t, \mathcal{F}, r$)

1: **Input**: $f, \mathbf{x}, t, \mathcal{F}, r$
2: Generate $\mathbf{u}_0$ randomly from $\mathbf{S}_r^d$
3: **for** $\tau = 0, \ldots, t$ **do**
4:    $\mathbf{u}_{\tau+1} = \mathbf{u}_\tau - \eta(\nabla f(\mathbf{x} + \mathbf{u}_\tau) - \nabla f(\mathbf{x}))$
5: **end for**
6: **if** $\min_{i \in [t+1], \|\mathbf{u}_i\| \leq U} \hat{f}_{\mathbf{x}}(\mathbf{u}_i) \leq -2.5\mathcal{F}$
7:    **return** $\mathbf{u}_{\tau'}, \tau' = \arg\min_{i \in [t+1], \|\mathbf{u}_i\| \leq U} \hat{f}_{\mathbf{x}}(\mathbf{u}_i)$
8: **else return** 0

---

**Algorithm 3** NCFind ($\mathbf{y}_{0:\tau}, \mathbf{u}_{0:\tau}$)

1: **if** $\min_{j=0,\ldots,\tau} \|\mathbf{y}_j - \mathbf{u}_j\| \geq \zeta\sqrt{6\eta\mathcal{F}}$
2:    **return** $\mathbf{y}_j, j = \min\{j' : \|\mathbf{y}_{j'} - \mathbf{u}_{j'}\| \geq \zeta\sqrt{6\eta\mathcal{F}}\}$

3: **else return** $\mathbf{y}_\tau - \mathbf{u}_\tau$

---

**Algorithm 2** NEON$^+$($f, \mathbf{x}, t, \mathcal{F}, U, \zeta, r$)

1: **Input**: $f, \mathbf{x}, t, \mathcal{F}, U, \zeta, r$
2: Generate $\mathbf{y}_0 = \mathbf{u}_0$ randomly from $\mathbf{S}_r$
3: **for** $\tau = 0, \ldots, t$ **do**
4:    **if** $\Delta_{\mathbf{x}}(\mathbf{y}_\tau, \mathbf{u}_\tau) < -\frac{\gamma}{2}\|\mathbf{y}_\tau - \mathbf{u}_\tau\|^2$
      **then**
5:       **return** $\mathbf{v} =$ NCFind($\mathbf{y}_{0:\tau}, \mathbf{u}_{0:\tau}$)
6:    **end if**
7:    compute $(\mathbf{y}_{\tau+1}, \mathbf{u}_{\tau+1})$ by (8)
8: **end for**
9: **if** $\min_{i, \|\mathbf{y}_i\| \leq U} \hat{f}_{\mathbf{x}}(\mathbf{y}_i) \leq -2\mathcal{F}$ **then**
10:    let $\tau' = \arg\min_{i, \|\mathbf{y}_i\| \leq U} \hat{f}_{\mathbf{x}}(\mathbf{y}_i)$
11:    **return** $\mathbf{y}_{\tau'}$
12: **else**
13:    **return** 0
14: **end if**

---

Hessian $\nabla^2 f(\mathbf{x})$. To provide a formal analysis, we will first analyze the following recurrence:

$$\mathbf{u}_\tau = \mathbf{u}_{\tau-1} - \eta(\nabla f(\mathbf{x} + \mathbf{u}_{\tau-1}) - \nabla f(\mathbf{x})), \tau = 1, \ldots \tag{6}$$

starting with a random noise vector $\mathbf{u}_0$, which is drawn from the sphere of an Euclidean ball with a proper radius $r$ denoted by $\mathbf{S}_r^d$. It is notable that the recurrence in (6) is slightly different from that in (5). We emphasize that this simple change is useful for *extracting the NC at any points* whose Hessian has a negative eigen-value not just at non-degenerate saddle points, which can be used in some stochastic or deterministic algorithms [1, 4, 21, 16]. The proposed procedure NEON based on the above sequence for finding a NC direction of $\nabla^2 f(\mathbf{x})$ is presented in Algorithm 1, where $\hat{f}_{\mathbf{x}}(\mathbf{u})$ is defined in (7). The following theorem states our result of NEON for extracting the NC.

**Theorem 1.** *Under Assumption 1 (i), let $\gamma \in (0,1)$ and $\delta \in (0,1)$ be a sufficiently small. For any constant $\hat{c} \geq 18$, there exists a constant $c_{\max}$ that depends on $\hat{c}$, such that if NEON is called with $t = \hat{c}\frac{\log(dL_1/(\gamma\delta))}{\eta\gamma}$, $\mathcal{F} = \eta\gamma^3 L_1 L_2^{-2} \log^{-3}(dL_1/(\gamma\delta))$, $r = \sqrt{\eta}\gamma^2 L_1^{-1/2} L_2^{-1} \log^{-2}(dL_1/(\gamma\delta))$, $U = 4\hat{c}(\sqrt{\eta L_1}\mathcal{F}/L_2)^{1/3}$ and a constant $\eta \leq c_{\max}/L_1$, then at a point $\mathbf{x}$ satisfying $\lambda_{\min}(\nabla^2 f(\mathbf{x})) \leq -\gamma$ with high probability $1 - \delta$ it returns $\mathbf{u}$ such that $\frac{\mathbf{u}^\top \nabla^2 f(\mathbf{x})\mathbf{u}}{\|\mathbf{u}\|^2} \leq -\frac{\gamma}{8\hat{c}^2 \log(dL_1/(\gamma\delta))} \leq -\tilde{\Omega}(\gamma)$. If NEON returns $\mathbf{u} \neq 0$, then the above inequality must hold; if NEON returns 0, we can conclude that $\lambda_{\min}(\nabla^2 f(\mathbf{x})) \geq -\gamma$ with high probability $1 - O(\delta)$.*

**Remark:** The above theorem shows that *at any point $\mathbf{x}$ whose Hessian has a negative eigen-value* (including non-degenerate saddle points), NEON can find a NC of $\nabla^2 f(\mathbf{x})$ with high probability.

## 4.2 Finding NC by Accelerated Gradient Method

Although NEON provides a similar guarantee for extracting a NC as that provided by the Power method, but its iteration complexity $O(1/\gamma)$ is worse than that of the Lanczos method, i.e., $O(1/\sqrt{\gamma})$. In this subsection, we present a first-order method that matches $O(1/\sqrt{\gamma})$ of the Lanczos method.

Let us recall the sequence (6), which is essentially an application of gradient descent (GD) method to the following objective function:

$$\hat{f}_{\mathbf{x}}(\mathbf{u}) = f(\mathbf{x} + \mathbf{u}) - f(\mathbf{x}) - \nabla f(\mathbf{x})^\top \mathbf{u}. \tag{7}$$

In the sequel, we write $\hat{f}_{\mathbf{x}}(\mathbf{u}) = \hat{f}(\mathbf{u})$, where the dependent $\mathbf{x}$ should be clear from the context. By the Lipschitz continuous Hessian condition, we have that

$$\frac{1}{2}\mathbf{u}^\top \nabla^2 f(\mathbf{x})\mathbf{u} - \frac{L_2}{6}\|\mathbf{u}\|^3 \leq \hat{f}(\mathbf{u}).$$

It implies that if $\hat{f}(\mathbf{u})$ is sufficiently less than zero and $\|\mathbf{u}\|$ is not too large, then $\frac{\mathbf{u}^\top \nabla^2 f(\mathbf{x})\mathbf{u}}{\|\mathbf{u}\|^2}$ will be sufficiently less than zero. Hence, NEON can be explained as using GD updates to decrease $\hat{f}(\mathbf{u})$.

A natural question to ask is whether the convergence of GD updates of NEON can be accelerated by accelerated gradient (AG) methods. It is well-known from convex optimization literature that AG methods can accelerate the convergence of GD method for smooth problems. Recently, several studies have explored AG methods for non-convex optimization [15, 19, 3, 12]. Notably, [19] analyzed the behavior of AG methods near strict saddle points and investigated the rate of divergence from a strict saddle point for toy quadratic problems. [12] analyzed a single-loop algorithm based on Nesterov's AG method for deterministic non-convex optimization. However, none of these studies provide an explicit complexity guarantee on extracting NC from the Hessian matrix for a general non-convex problem. Inspired by these studies, we will show that Nesterov's AG (NAG) method [18] when applied the function $\hat{f}(\mathbf{u})$ can find a NC with a complexity of $\widetilde{O}(1/\sqrt{\gamma})$.

The updates of NAG method applied to the function $\hat{f}(\mathbf{u})$ at a given point $\mathbf{x}$ is given by

$$\begin{aligned}
\mathbf{y}_{\tau+1} &= \mathbf{u}_\tau - \eta \nabla \hat{f}(\mathbf{u}_\tau), \\
\mathbf{u}_{\tau+1} &= \mathbf{y}_{\tau+1} + \zeta(\mathbf{y}_{\tau+1} - \mathbf{y}_\tau),
\end{aligned} \tag{8}$$

where $\zeta(\mathbf{y}_{\tau+1} - \mathbf{y}_\tau)$ is the momentum term, and $\zeta \in (0,1)$ is the momentum parameter. The proposed algorithm based on the NAG method (referred to as NEON$^+$) for extracting NC of a Hessian matrix $\nabla^2 f(\mathbf{x})$ is presented in Algorithm 2, where

$$\Delta_{\mathbf{x}}(\mathbf{y}_\tau, \mathbf{u}_\tau) = \hat{f}_{\mathbf{x}}(\mathbf{y}_\tau) - \hat{f}_{\mathbf{x}}(\mathbf{u}_\tau) - \nabla \hat{f}_{\mathbf{x}}(\mathbf{u}_\tau)^\top (\mathbf{y}_\tau - \mathbf{u}_\tau),$$

and NCFind is a procedure that returns a NC by searching over the history $\mathbf{y}_{0:\tau}, \mathbf{u}_{0:\tau}$ shown in Algorithm 3. The condition check in Step 4 is to detect easy cases such that NCFind can easily find a NC in historical solutions without continuing the update, which is designed following a similar procedure called Negative Curvature Exploitation (NCE) proposed in [12]. However, the difference is that NCFind is tailored to finding a negative curvature satisfying (3), while NCE in [12] is for ensuring a decrease on a modified objective. The theoretical result of NEON$^+$ is presented below.

**Theorem 2.** *Under Assumption 1 (i), let $\gamma \in (0,1)$ and $\delta \in (0,1)$ be a sufficiently small. For any constant $\hat{c} \geq 43$, there exists a constant $c_{\max}$ that depends on $\hat{c}$, such that if NEON$^+$ is called with $t = \sqrt{\frac{\hat{c}\log(dL_1/(\gamma\delta))}{\eta\gamma}}$, $\mathcal{F} = \eta\gamma^3 L_1 L_2^{-2} \log^{-3}(dL_1/(\gamma\delta))$, $r = \sqrt{\eta}\gamma^2 L_1^{-1/2} L_2^{-1} \log^{-2}(dL_1/(\gamma\delta))$, $U = 12\hat{c}(\sqrt{\eta L_1}\mathcal{F}/L_2)^{1/3}$, a small constant $\eta \leq c_{\max}/L_1$, and a momentum parameter $\zeta = 1 - \sqrt{\eta\gamma}$, then at any point $\mathbf{x}$ satisfying $\lambda_{\min}(\nabla^2 f(\mathbf{x})) \leq -\gamma$ with high probability $1 - \delta$ it returns $\mathbf{u}$ such that $\frac{\mathbf{u}^\top \nabla^2 f(\mathbf{x})\mathbf{u}}{\|\mathbf{u}\|^2} \leq -\frac{\gamma}{72\hat{c}^2 \log(dL_1/(\gamma\delta))} \leq -\widetilde{\Omega}(\gamma)$. If NEON$^+$ returns $\mathbf{u} \neq 0$, then the above inequality must hold; if NEON$^+$ returns $0$, we can conclude that $\lambda_{\min}(\nabla^2 f(\mathbf{x})) \geq -\gamma$ with high probability $1 - O(\delta)$.*

### 4.3 Stochastic Approach for Extracting NC

In this subsection, we present a stochastic approach for extracting NC for $F(\mathbf{x})$ in (1). For simplicity, we refer to both NEON and NEON$^+$ as NEON. The challenge in employing NEON for finding a NC for the original function $F(\mathbf{x})$ in (1) is that we cannot evaluate the gradient of $F(\mathbf{x})$ exactly. To address this issue, we resort to the mini-batching technique.

Let $\mathcal{S} = \{\xi_1, \ldots, \xi_m\}$ denote a set of random samples and define a sub-sampled function $F_{\mathcal{S}}(\mathbf{x}) = \frac{1}{|\mathcal{S}|} \sum_{\xi \in \mathcal{S}} f(\mathbf{x}; \xi)$. Then we apply NEON to $F_{\mathcal{S}}(\mathbf{x})$ for finding an approximate NC $\mathbf{u}_{\mathcal{S}}$ of $\nabla^2 F_{\mathcal{S}}(\mathbf{x})$. Below, we show that as long as $m$ is sufficiently large, $\mathbf{u}_{\mathcal{S}}$ is also an approximate NC of $\nabla^2 F(\mathbf{x})$.

**Theorem 3.** *Under Assumption 1 (i), for a sufficiently small $\delta \in (0,1)$ and $\hat{c} \geq 43$, let $m \geq \frac{16L_1^2 \log(6d/\delta)}{s^2\gamma^2}$, where $s = \frac{\log^{-1}(3dL_1/(2\gamma\delta))}{(12\hat{c})^2}$ is a proper small constant. If $\lambda_{\min}(\nabla^2 F(\mathbf{x})) \leq -\gamma$, there exists $c > 0$ such that with probability $1 - \delta$, NEON$(F_{\mathcal{S}}, \mathbf{x}, t, \mathcal{F}, r)$ returns a vector $\mathbf{u}_{\mathcal{S}}$ such that $\frac{\mathbf{u}_{\mathcal{S}}^\top \nabla^2 F(\mathbf{x})\mathbf{u}_{\mathcal{S}}}{\|\mathbf{u}_{\mathcal{S}}\|^2} \leq -c\gamma$, where $c = (12\hat{c})^{-2} \log^{-1}(3dL_1/(2\gamma\delta))$. If NEON$(F_{\mathcal{S}}, \mathbf{x}, t, \mathcal{F}, r)$ returns $0$, then with high probability $1 - O(\delta)$ we have $\lambda_{\min}(\nabla^2 F(\mathbf{x})) \geq -2\gamma$. In either case, NEON terminates with an IFO complexity of $\widetilde{O}(1/\gamma^3)$ or $\widetilde{O}(1/\gamma^{2.5})$ corresponding to Algorithm 1 and Algorithm 2, respectively.*

| **Algorithm 4** NEON-$\mathcal{A}$ | **Algorithm 6** SCSG-epoch: $(\mathbf{x}, \mathcal{S}_1, b)$ |
|---|---|
| 1: **Input**: $\mathbf{x}_1$, other parameters of algorithm $\mathcal{A}$<br>2: **for** $j = 1, 2, \ldots,$ **do**<br>3:    Compute $(\mathbf{y}_j, \mathbf{z}_j) = \mathcal{A}(\mathbf{x}_j)$<br>4:    **if** first-order condition of $\mathbf{y}_j$ not met **then**<br>5:       let $\mathbf{x}_{j+1} = \mathbf{z}_j$<br>6:    **else**<br>7:       $\mathbf{u}_j = \text{NEON}(F_{\mathcal{S}_2}, \mathbf{y}_j, t, \mathcal{F}, r)$<br>8:       **if** $\mathbf{u}_j = 0$ **return** $\mathbf{y}_j$<br>9:       **else** let $\mathbf{x}_{j+1} = \mathbf{y}_j - \frac{c\gamma\bar{\xi}}{L_2}\frac{\mathbf{u}_j}{\|\mathbf{u}_j\|}$<br>10:    **end if**<br>11: **end for** | 1: **Input**: $\mathbf{x}$, an independent set of samples $\mathcal{S}_1$ and $b \le |\mathcal{S}_1|$<br>2: Set $m_1 = |\mathcal{S}_1|$, $\eta = c'(m_1/b)^{-2/3}$, $c' \le 1/6$<br>3: Compute $\nabla F_{\mathcal{S}}(\mathbf{x}_{j-1})$ and let $\mathbf{x}_0 = \mathbf{x}$<br>4: Generate $N \sim \text{Geom}(m_1/(m_1 + b))$<br>5: **for** $k = 1, 2, \ldots, N$ **do**<br>6:    Sample samples $\mathcal{S}_k$ of size $b$<br>7:    $\mathbf{v}_k = \nabla F_{\mathcal{S}_k}(\mathbf{x}_{k-1}) - \nabla F_{\mathcal{S}_k}(\mathbf{x}_0) + \nabla F_{\mathcal{S}}(\mathbf{x}_0)$<br>8:    $\mathbf{x}_k = \mathbf{x}_{k-1} - \eta\mathbf{v}_k$<br>9: **end for**<br>10: **return** $\mathbf{x}_N$ |

# 5  First-order Algorithms for Stochastic Non-Convex Optimization

In this section, we will first describe a general framework for promoting existing first-order stochastic algorithms denoted by $\mathcal{A}$ to enjoy a second-order convergence, which is shown in Algorithm 4. Here, we require $\mathcal{A}(\mathbf{x}_j)$ to return two points $(\mathbf{y}_j, \mathbf{z}_j)$ that satisfy (9) and the mini-batch sample size $m = |\mathcal{S}_2|$ satisfies the condition in Lemma 3. The proposed NEON is used for escaping from a saddle point. It should be noted that Algorithm 4 is abstract depending on how to implement Step 3, how to check the first-order condition, and how to set the step size parameter $\bar{\xi}$ in Step 9.

For theoretical interest, we will analyze Algorithm 4 with a Rademacher random variable $\bar{\xi} \in \{1, -1\}$ and its three main components satisfying the following properties.

**Property 1.** *(1) Step 7 - Step 9 guarantees that if $\lambda_{\min}(\nabla^2 F(\mathbf{y}_j)) \le -\gamma$, there exists $C > 0$ such that $\mathrm{E}[F(\mathbf{x}_{j+1}) - F(\mathbf{y}_j)] \le -C\gamma^3$. Let the total IFO complexity of Step 7 - Step 9 be $T_n$. (2) There exists a first-order stochastic algorithm $(\mathbf{y}_j, \mathbf{z}_j) = \mathcal{A}(\mathbf{x}_j)$ that satisfies:*

$$
\begin{aligned}
&\text{if } \|\nabla F(\mathbf{y}_j)\| \ge \epsilon, \text{then } \mathrm{E}[F(\mathbf{z}_j) - F(\mathbf{x}_j)] \le -\varepsilon(\epsilon, \alpha) \\
&\text{if } \|\nabla F(\mathbf{y}_j)\| \le \epsilon, \text{then } \mathrm{E}[F(\mathbf{y}_j) - F(\mathbf{x}_j)] \le C\gamma^3/2
\end{aligned}
\tag{9}
$$

*where $\varepsilon(\epsilon, \alpha)$ is a function of $\epsilon$ and a parameter $\alpha > 0$. Let the total IFO complexity of $\mathcal{A}(\mathbf{x})$ be $T_a$. (3) the check of first-order condition can be implemented by using a mini-batch of samples $\mathcal{S}$, i.e., $\|\nabla F_{\mathcal{S}}(\mathbf{y}_j)\| \le \epsilon$, where $\mathcal{S}$ is independent of $\mathbf{y}_j$ such that $\|\nabla F(\mathbf{y}_j) - \nabla F_{\mathcal{S}}(\mathbf{y}_j)\| \le \epsilon/2$. Let the IFO complexity of checking the first-order condition be $T_c$.*

Property (1) can be guaranteed by Theorem 3 and Lemma 1. When using NEON, $T_n = \widetilde{O}(1/\gamma^3)$ and when using NEON$^+$, $T_n = \widetilde{O}(1/\gamma^{2.5})$. For Property (2), we will analyze several interesting algorithms. Property (3) can be guaranteed by Lemma 2 in the supplement under Assumption (1) (iii) with $T_c = \widetilde{O}(\frac{1}{\epsilon^2})$. Based on the above properties, we have the following convergence of Algorithm 4.

**Theorem 4.** *Assume Properties 1 hold. Then with high probability $1 - \delta$, NEON-$\mathcal{A}$ terminates with a total IFO complexity of $\widetilde{O}(\max(\frac{1}{\varepsilon(\epsilon, \alpha)}, \frac{1}{\gamma^3})(T_n + T_a + T_c))$. Upon termination, with high probability $\|\nabla F(\mathbf{y}_j)\| \le O(\epsilon)$ and $\lambda_{\min}(\nabla^2 F(\mathbf{y}_j)) \ge -2\gamma$, where $\widetilde{O}(\cdot)$ hides logarithmic factors of $d$ and $1/\delta$, and problem's other constant parameters.*

Next, we present corollaries of Theorem 4 for several instances of $\mathcal{A}$, including stochastic gradient descent (SGD) method, stochastic momentum (SM) methods, mini-batch SGD (MSGD), and SCSG. SGD and its momentum variants (including stochastic heavy-ball (SHB) method and stochastic Nesterov's accelerated gradient (SNAG) method) are popular stochastic algorithms for solving a stochastic non-convex optimization problem. We will consider them in a unified framework as established in [22]. The updates of SM starting from $\mathbf{x}_0$ are

$$
\begin{aligned}
\widehat{\mathbf{x}}_{\tau+1} &= \mathbf{x}_\tau - \eta\nabla f(\mathbf{x}_\tau; \xi_\tau), \\
\widehat{\mathbf{x}}_{\tau+1}^s &= \mathbf{x}_\tau - s\eta\nabla f(\mathbf{x}_\tau; \xi_\tau), \\
\mathbf{x}_{\tau+1} &= \widehat{\mathbf{x}}_{\tau+1} + \beta(\widehat{\mathbf{x}}_{\tau+1}^s - \widehat{\mathbf{x}}_\tau^s),
\end{aligned}
\tag{10}
$$

| **Algorithm 5** SM: $(\mathbf{x}_0, \eta, \beta, s, t)$ |
| --- |
| 1: **for** $\tau = 0, 1, 2, \ldots, t$ **do** |
| 2:     Compute $\mathbf{x}_{\tau+1}$ according to (10) |
| 3:     Compute $\mathbf{x}_{\tau+1}^+$ according to (11) |
| 4: **end for** |
| 5: **return** $(\mathbf{x}_{\tau'}^+, \mathbf{x}_{t+1}^+)$, where $\tau' \in \{0, \ldots, t\}$ is a randomly generated. |

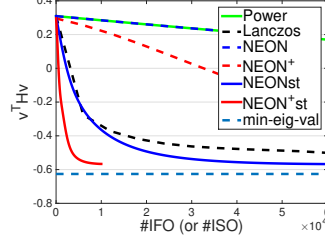

Figure 1: NEON vs Second-order Methods for Extracting NC

for $\tau = 0, \ldots, t$ and $\widehat{\mathbf{x}}_0^s = \mathbf{x}_0$, where $\beta \in (0, 1)$ is a momentum constant, $\eta$ is a step size, $s = 0, 1, 1/(1 - \beta)$ corresponds to SHB, SNAG and SGD. Let sequence $\mathbf{x}_\tau^+$ with $\mathbf{x}_0^+ = \mathbf{x}_0$ be defined as

$$\mathbf{x}_\tau^+ = \mathbf{x}_\tau + \mathbf{p}_\tau, \tau \geq 1, \quad \mathbf{p}_\tau = \frac{\beta}{1 - \beta}(\mathbf{x}_\tau - \mathbf{x}_{\tau-1} - s\eta\nabla f(\mathbf{x}_{\tau-1}; \xi_{\tau-1})). \tag{11}$$

We can implement $\mathcal{A}$ by Algorithm 5 and have the following result.

**Corollary 5.** *Let $\mathcal{A}(\mathbf{x}_j)$ be implemented by Algorithm 5 with $t = \Theta(1/\epsilon^2)$ iterations, $\eta = \Theta(\epsilon^2), \beta \in (0, 1), s \in (0, 1/(1 - \beta))$. Then $T_a = O(1/\epsilon^2)$ and $\varepsilon(\epsilon, \alpha) = \Theta(\epsilon^2)$. Suppose that $\gamma \geq \epsilon^{2/3}$ and $\mathrm{E}[\|\nabla f(\mathbf{x}; \xi)\|^2]$ is bounded for $s \neq 1/(1 - \beta)$. Then with high probability, NEON-SM finds an $(\epsilon, \gamma)$-SPP with a total IFO complexity of $\widetilde{O}(\max(\frac{1}{\epsilon^2}, \frac{1}{\gamma^3})(T_n + \frac{1}{\epsilon^2}))$, where $T_n = \widetilde{O}(1/\gamma^3)$ (NEON) or $T_n = \widetilde{O}(1/\gamma^{2.5})$ (NEON$^+$).*

**Remark:** When $\gamma = \epsilon^{1/2}$, NEON-SM has an IFO complexity of $\widetilde{O}(\frac{1}{\epsilon^4})$.

MSGD computes $(\mathbf{y}_j, \mathbf{z}_j)$ by

$$\mathbf{z}_j = \mathbf{x}_j - L_1^{-1}\nabla F_{\mathcal{S}_1}(\mathbf{x}_j), \quad \mathbf{y}_j = \mathbf{x}_j \tag{12}$$

where $\mathcal{S}_1$ is a set of samples independent of $\mathbf{x}_j$.

**Corollary 6.** *Let $\mathcal{A}(\mathbf{x}_j)$ be implemented by (12) with $|\mathcal{S}_1| = \widetilde{O}(1/\epsilon^2)$. Then $T_a = \widetilde{O}(1/\epsilon^2)$ and $\varepsilon(\epsilon, \alpha) = \frac{\epsilon^2}{4L_1}$. With high probability, NEON-MSGD finds an $(\epsilon, \gamma)$-SPP with a total IFO complexity of $\widetilde{O}(\max(\frac{1}{\epsilon^2}, \frac{1}{\gamma^3})(T_n + 1/\epsilon^2))$.*

**Remark:** Compared to Corollary 5, there is no requirement on $\gamma \geq \epsilon^{2/3}$, which is due to that MSGD can guarantee that $\mathrm{E}[F(\mathbf{y}_j) - F(\mathbf{x}_j)] \leq 0$.

SCSG was proposed in [14], which only provides a first-order convergence guarantee. SCSG runs with multiple epochs, and each epoch uses similar updates as SVRG with three distinct features: (i) it was applied to a sub-sampled function $F_{\mathcal{S}_1}$; (ii) it allows for using a mini-batch samples of size $b$ independent of $\mathcal{S}_1$ to compute stochastic gradients; (ii) the number of updates of each epoch is a random number following a geometric distribution dependent on $b$ and $|\mathcal{S}_1|$. These features make each SGCG epoch denoted by SCSG-epoch$(\mathbf{x}, \mathcal{S}_1, b)$ have an expected IFO complexity of $T_a = O(|\mathcal{S}_1|)$. We present SCSG-epoch$(\mathbf{x}, \mathcal{S}_1, b)$ in Algorithm 6. For using SCSG, $\mathbf{y}_j$ and $\mathbf{z}_j$ are

$$\mathbf{y}_j = \text{SCSG-epoch}(\mathbf{x}_j, \mathcal{S}_1, b), \quad \mathbf{z}_j = \mathbf{y}_j \tag{13}$$

**Corollary 7.** *Let $\mathcal{A}(\mathbf{x}_j)$ be implemented by (13) with $|\mathcal{S}_1| = \widetilde{O}\left(\max(1/\epsilon^2, 1/(\gamma^{9/2}b^{1/2}))\right)$. Then $\varepsilon(\epsilon, \alpha) = \Omega(\epsilon^{4/3}/b^{1/3})$ and $\mathrm{E}[T_a] = \widetilde{O}\left(\max(1/\epsilon^2, 1/(\gamma^{9/2}b^{1/2}))\right)$. With high probability, NEON-SCSG finds an $(\epsilon, \gamma)$-SSP with an expected total IFO complexity of $\widetilde{O}(\max(\frac{b^{1/3}}{\epsilon^{4/3}}, \frac{1}{\gamma^3})(T_n + 1/\epsilon^2 + 1/(\gamma^{9/2}b^{1/2})))$, where $T_n = \widetilde{O}(1/\gamma^3)$ (NEON) or $T_n = \widetilde{O}(1/\gamma^{2.5})$ (NEON$^+$).*

**Remark:** When $\gamma = \epsilon^{1/2}, b = 1/\sqrt{\epsilon}$, NEON-SCSG has an expected IFO complexity of $\widetilde{O}(\frac{1}{\epsilon^{3.5}})$. When $\gamma \geq \epsilon^{4/9}, b = 1$, NEON-SCSG has an expected IFO complexity of $\widetilde{O}(1/\epsilon^{3.33})$.

Finally, we mention that the proposed NEON or NEON$^+$ can be used in existing second-order stochastic algorithms that require a NC direction as a substitute of second-order methods [1, 20]. [1] developed Natasha2, which uses second-order online Oja's algorithm for finding the NC. [20] developed a stochastic algorithm for solving a finite-sum problem by using SVRG and a second-order stochastic algorithm for computing the NC. We can replace the second-order methods for computing a NC in these algorithms by the proposed NEON or NEON$^+$, with the resulting algorithms referred to as NEON-Natasha and NEON-SVRG. It is a simple exercise to derive the convergence results in Table 1, which is left to interested readers.

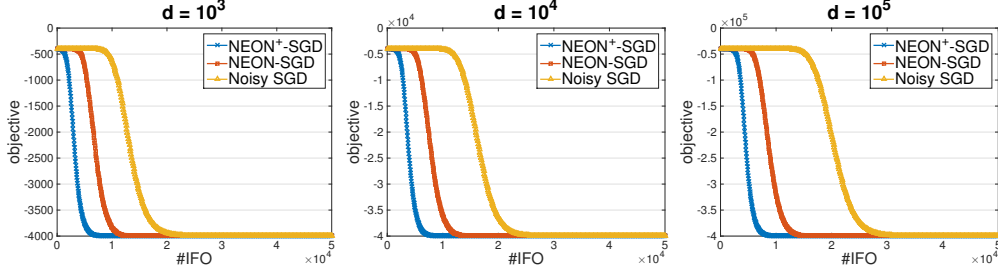

Figure 2: NEON-SGD vs Noisy SGD. (All algorithms converge to local minimum)

# 6 Experiments

**Extracting NC.** First, we present some simulations to verify the proposed NEON procedures for extracting NC. To this end, we consider minimizing non-linear least square loss with a non-convex regularizer for classification, i.e., $F(\mathbf{x}) = \sum_{i=1}^{d} \frac{x_i^2}{1+x_i^2} + \frac{\lambda}{n} \sum_{i=1}^{n} (b_i - \sigma(\mathbf{x}^\top \mathbf{a}_i))^2$, where $b_i \in \{0, 1\}$ denotes the label and $\mathbf{a}_i \in \mathbb{R}^d$ denotes the feature vector of the $i$-th data, $\lambda > 0$ is a trade-off parameter, and $\sigma(\cdot)$ is a sigmoid function. We generate a random vector $\mathbf{x} \sim \mathcal{N}(0, I)$ as the target point to construct $\hat{F}_{\mathbf{x}}(\mathbf{u})$ and compute a NC of $\nabla^2 F(\mathbf{x})$. We use a binary classification data named gisette from the libsvm data website that has $n = 6000$ examples and $d = 5000$ features, and set $\lambda = 3$ in our simulation to ensure there is significant NC from the non-linear least-square loss. The step size $\eta$ and initial radius in NEON procedures are set to be 0.01 and the momentum parameter in NEON$^+$ is set to be 0.9. These values are tuned in a certain range.

We compare the two NEON procedures and their stochastic variants (denoted by NEON-st and NEON$^+$-st in the figure) with second-order methods that use HVPs, namely the Power method and the Lanczos method, where the HVPs are calculated exactly. The result is shown in Figure 1 whose $y$-axis denotes the value of $\widehat{\mathbf{u}}^\top H \widehat{\mathbf{u}}$, where $\widehat{\mathbf{u}}$ represents the found normalized NC vector and $H = \nabla^2 F(\mathbf{x})$ is the Hessian matrix. For NEON-st and NEON$^+$-st, we use a sample size of 100. Please note that the solid red curve corresponding to NEON$^+$-st terminates earlier due to that NCFind is executed. Several observations follow: (i) NEON performs similarly to the Power method (the two curves overlap in the figure); (ii) NEON$^+$ has a faster convergence than NEON; (iv) the stochastic versions of NEON and NEON$^+$ can quickly find a good NC directions than their full versions in terms of IFO complexity and are even competitive with the Lanczos method. We include several more results in the supplement.

**Escaping Saddles.** Second, we present some simulations to verify the proposed NEON and NEON$^+$ based algorithms for minimizing a stochastic objective. We consider a non-convex optimization problem with $f(\mathbf{x}; \xi) = \sum_{i=1}^{d} \xi_i (x_i^4 - 4x_i^2)$ where $\xi_i$ are a normal random variables with mean of 1 so that the saddle points of the expected function are known [10]. Assuming the noise $\xi$ is only accessed through a sampler, then we compare NEON-SGD with a state-of-the-art algorithm Noisy SGD [5] for different values of $d \in \{10^3, 10^4, 10^5\}$. The step size of Noisy SGD is tuned in a wide range and the best one is used. The step size in NEON procedures are set to be the same value as Noisy SGD. The radius in NEON procedures is set to be 0.01 and the momentum paramenter in NEON$^+$ is set to be 0.9. The mini-batch size is tuned from $\{50, 100, 200, 500\}$. All algorithms are started with a same saddle point as the initial solution. The results are presented in Figure 2, showing that two variants of NEON-SGD methods can escape saddles faster than Noisy SGD. NEON$^+$-SGD escapes saddle points the fastest among all algorithms for different values of $d$. In addition, the increasing of dimensionality $d$ has much larger effect on the IFO complexity of Noisy-SGD than that of NEON-SGD methods, which is consistent with theoretical results.

# 7 Conclusions

We have proposed novel first-order procedures to extract negative curvature from a Hessian matrix by using a noise-initiated sequence, which are of independent interest. A general framework for promoting a first-order stochastic algorithm to enjoy a second-order convergence is also proposed. Based on the proposed general framework, we designed several first-order stochastic algorithms with state-of-the-art second-order convergence guarantee.

## Acknowledgement

The authors thank the anonymous reviewers for their helpful comments. Y. Xu and T. Yang are partially supported by National Science Foundation (IIS-1545995).

## Footnotes

[1] We abuse the same notation $f$ here.

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
