[Supplementary Material · neon-with-supplementary.pdf]

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

# Supplementary Material

## A Proof of Lemma 1

*Proof.* Let $\eta = \frac{c\gamma}{L_2}\text{sign}(\mathbf{v}^\top \nabla f(\mathbf{x}))$ be the step size, so that $\mathbf{x}_+ = \mathbf{x} - \eta \mathbf{v}$. By the $L_2$-Lipschitz continuous Hessian of $f(\mathbf{x})$, we have

$$|f(\mathbf{x}_+) - f(\mathbf{x}) + \eta \mathbf{v}^\top \nabla f(\mathbf{x}) - \frac{1}{2}\eta^2 \mathbf{v}^\top \nabla^2 f(\mathbf{x})\mathbf{v}| \leq \frac{L_2}{6}\|\eta \mathbf{v}\|^3.$$

By noting that $\eta \mathbf{v}^\top \nabla f(\mathbf{x}) \geq 0$, we have

$$f(\mathbf{x}) - f(\mathbf{x}_+) \geq \eta \mathbf{v}^\top \nabla f(\mathbf{x}) - \frac{1}{2}\eta^2 \mathbf{v}^\top \nabla^2 f(\mathbf{x})\mathbf{v} - \frac{L_2}{6}\|\eta \mathbf{v}\|^3$$

$$\geq \frac{c^3\gamma^3}{2L_2^2} - \frac{c^3\gamma^3}{6L_2^2} = \frac{c^3\gamma^3}{3L_2^2},$$

where the last inequality uses that $\mathbf{v}^\top \nabla^2 f(\mathbf{x})\mathbf{v} \leq -c\gamma$, $\|\mathbf{v}\| = 1$, and the definition of $\eta$.

For the case of $\mathbf{x}'_+$, we have $\mathbf{x}'_+ = \mathbf{x} - \eta \mathbf{v}$, where $\eta = \bar{\xi}\frac{c\gamma}{L_2}$. Similarly, we can prove the expectation result of $\mathbf{x}'_+$. Then

$$\mathrm{E}[f(\mathbf{x}) - f(\mathbf{x}_+)] \geq \mathrm{E}[\eta \mathbf{v}^\top \nabla f(\mathbf{x}) - \frac{1}{2}\eta^2 \mathbf{v}^\top \nabla^2 f(\mathbf{x})\mathbf{v} - \frac{L_2}{6}\|\eta \mathbf{v}\|^3]$$

$$\geq \frac{c^3\gamma^3}{2L_2^2} - \frac{c^3\gamma^3}{6L_2^2} = \frac{c^3\gamma^3}{3L_2^2},$$

where we use $\mathrm{E}[\eta] = 0$, $\mathrm{E}[\eta^2] = \frac{c^2\gamma^2}{L_2^2}$, $\mathrm{E}[|\eta|^3] = \frac{c^3\gamma^3}{L_2^3}$, and $\|\mathbf{v}\| = 1$. $\qquad\square$

## B Concentration inequalities

We first present some concentration inequalities of random vectors and random matrices. Below, we let $\mathcal{S}_1$ and $\mathcal{S}_2$ to denote a set of random samples that are generated independently of $\mathbf{x}$.

**Lemma 2** ([8], Lemma 4). *Suppose Assumption 1(iii) holds. Let $\nabla F_{\mathcal{S}_1}(\mathbf{x}) = \frac{1}{|\mathcal{S}_1|}\sum_{\mathbf{z}_i \in \mathcal{S}_1}\nabla f(\mathbf{x}; \mathbf{z}_i)$. For any $\epsilon, \delta \in (0, 1)$, $\mathbf{x} \in \mathbb{R}^d$ when $|\mathcal{S}_1| \geq \frac{2G^2(1+8\log(1/\delta))}{\epsilon^2}$, we have $\mathrm{Pr}(\|\nabla F_{\mathcal{S}_1}(\mathbf{x}) - \nabla F(\mathbf{x})\| \leq \epsilon) \geq 1 - \delta$.*

**Lemma 3** ([22], Lemma 4). *Suppose Assumption 1(i) holds. Let $\nabla^2 F_{\mathcal{S}_2}(\mathbf{x}) = \frac{1}{|\mathcal{S}_2|}\sum_{\mathbf{z}_i \in \mathcal{S}_2}\nabla^2 f(\mathbf{x}; \mathbf{z}_i)$. For any $\epsilon, \delta \in (0, 1), \mathbf{x} \in \mathbb{R}^d$, when $|\mathcal{S}_2| \geq \frac{16L_1^2\log(2d/\delta)}{\epsilon^2}$, we have $\mathrm{Pr}(\|\nabla^2 F_{\mathcal{S}_2}(\mathbf{x}) - \nabla^2 F(\mathbf{x})\|_2 \leq \epsilon) \geq 1 - \delta$.*

**Claim**. In the following analysis, when we say high probability, it means there is a probability $1 - \delta$ with a small enough $\delta < 0$. In many cases, we prove an inequality for one iteration with high probability, which implies the final result with high probability using union bound with a finite number of iterations. Instead of repeating this argument, we will simply assume this is done. We can always set the $\delta$ in the involved parameters (in the logarithmic part) small enough to make our argument fly.

## C Proof of Theorem 3

Due to that the proof of Theorem 1 and Theorem 2 is lengthy, we postpone them into the end of the supplement.

*Proof.* By using a matrix concentration inequality in Lemma 3, if $m \geq \frac{16L_1^2\log(6d/\delta)}{s^2\gamma^2}$, with probability $1 - \delta/3$ we have

$$\|\nabla^2 F(\mathbf{x}) - \nabla^2 F_{\mathcal{S}}(\mathbf{x})\|_2 \leq s\gamma.$$

Since $\|L_1 I - \nabla^2 F(\mathbf{x})\|_2 = L_1 - \lambda_{\min}(\nabla^2 F(\mathbf{x}))$ and $\|L_1 I - \nabla^2 F_{\mathcal{S}}(\mathbf{x})\|_2 = L_1 - \lambda_{\min}(\nabla^2 F_{\mathcal{S}}(\mathbf{x}))$, if $\lambda_{\min}(\nabla^2 F(\mathbf{x})) \le -\gamma$, with probability $1 - \delta/3$,

$$(L_1 - \lambda_{\min}(\nabla^2 F(\mathbf{x}))) - (L_1 - \lambda_{\min}(\nabla^2 F_{\mathcal{S}}(\mathbf{x}))) \le \|(L_1 I - \nabla^2 F(\mathbf{x})) - (L_1 I - \nabla^2 F_{\mathcal{S}}(\mathbf{x}))\|_2 \le s\gamma.$$

As a result, with probability $1 - \delta/3$, $\lambda_{\min}(\nabla^2 F_{\mathcal{S}}(\mathbf{x})) \le -\gamma + s\gamma \le -3\gamma/4$ with $s = \frac{\log^{-1}(3dL_1/(2\gamma\delta))}{(12\hat{c})^2} \le 1/4$.

(i) The NEON applied to $F_{\mathcal{S}}$ can generate $\mathbf{u}_{\mathcal{S}}$ with probability $1 - 2\delta/3$ (over randomness in $\mathcal{S}$ and NEON) such that

$$\frac{\mathbf{u}_{\mathcal{S}}^\top \nabla^2 F_{\mathcal{S}}(\mathbf{x}) \mathbf{u}_{\mathcal{S}}}{\|\mathbf{u}_{\mathcal{S}}\|^2} \le \frac{-2\mathcal{F}'}{(4\hat{c}\mathcal{P}')^2} = \frac{-\gamma}{8\hat{c}^2 \log(3dL_1/(2\gamma\delta))} \le \frac{-\gamma}{72\hat{c}^2 \log(3dL_1/(2\gamma\delta))} \le -\widetilde{\Omega}(\gamma),$$

where $\mathcal{F}' = \eta L_1 \gamma^3 L_2^{-2} \log^{-3}(3dL_1/(2\gamma\delta))$ and $\mathcal{P}' = \sqrt{\eta L_1} \gamma L_2^{-1} \log^{-1}(3dL_1/(2\gamma\delta))$.

(ii) Similarly, the NEON$^+$ applied to $F_{\mathcal{S}}$ can generate $\mathbf{u}_{\mathcal{S}}$ with probability $1 - 2\delta/3$ (over randomness in $\mathcal{S}$ and NEON$^+$) such that

$$\frac{\mathbf{u}_{\mathcal{S}}^\top \nabla^2 F_{\mathcal{S}}(\mathbf{x}) \mathbf{u}_{\mathcal{S}}}{\|\mathbf{u}_{\mathcal{S}}\|^2} \le \frac{-\gamma}{72\hat{c}^2 \log(3dL_1/(2\gamma\delta))} \le -\widetilde{\Omega}(\gamma).$$

As a result, both for NEON and NEON$^+$, with probability $1 - \delta$ (over randomness in $\mathcal{S}$ and NEON or NEON$^+$),

$$\left| \frac{\mathbf{u}_{\mathcal{S}}^\top \nabla^2 F(\mathbf{x}) \mathbf{u}_{\mathcal{S}}}{\|\mathbf{u}_{\mathcal{S}}\|^2} - \frac{\mathbf{u}_{\mathcal{S}}^\top \nabla^2 F_{\mathcal{S}}(\mathbf{x}) \mathbf{u}_{\mathcal{S}}}{\|\mathbf{u}_{\mathcal{S}}\|^2} \right| \le \|\nabla^2 F(\mathbf{x}) - \nabla^2 F_{\mathcal{S}}(\mathbf{x})\|_2 \le s\gamma.$$

Hence,

$$\frac{\mathbf{u}_{\mathcal{S}}^\top \nabla^2 F(\mathbf{x}) \mathbf{u}_{\mathcal{S}}}{\|\mathbf{u}_{\mathcal{S}}\|^2} \le -\frac{\gamma}{72\hat{c}^2 \log(3dL_1/(2\gamma\delta))} + s\gamma = -c\gamma,$$

where $s = \frac{\log^{-1}(3dL_1/(2\gamma\delta))}{(12\hat{c})^2}$. If NEON or NEON$^+$ returns 0, we then terminate the algorithm, which guarantees that $\lambda_{\min}(F_{\mathcal{S}}(\mathbf{x})) \ge -\gamma$ with high probability and therefore $-\lambda_{\min}(\nabla^2 F(\mathbf{x})) \le \gamma + s\gamma \le 2\gamma$ with high probability. $\quad\square$

# D  Proof of Theorem 4

*Proof.* To prove the convergence of the generic algorithm, we need to prove the total number of iterations NEON-$\mathcal{A}$ before termination. Upon termination, it then holds that $\|\nabla F_{\mathcal{S}_1}(\mathbf{y}_j)\| \le \epsilon$, $\lambda_{\min}(\nabla^2 F_{\mathcal{S}_2}(\mathbf{x}_j)) \ge -\gamma$. By concentration inequalities, we have $\lambda_{\min}(\nabla^2 F(\mathbf{y}_j)) \ge -2\gamma$ and $\|\nabla F(\mathbf{y}_j)\| \le O(\epsilon)$ hold with high probability. Before termination, let us consider two cases based on the first-order condition at point $\mathbf{y}_j$: (1) $\|\nabla F(\mathbf{y}_j)\| \ge \epsilon$ and (2) $\|\nabla F(\mathbf{y}_j)\| < \epsilon$.

For the first case, by (9) we have

$$\mathrm{E}[F(\mathbf{z}_j) - F(\mathbf{x}_j)] \le -\varepsilon(\epsilon, \alpha).$$

Since the the first-oder condition of $\mathbf{y}_j$ not met in this case, then $\mathbf{x}_{j+1} = \mathbf{z}_j$, so that

$$\mathrm{E}[F(\mathbf{x}_{j+1}) - F(\mathbf{x}_j)] \le -\varepsilon(\epsilon, \alpha). \tag{14}$$

For the second case, by (9) we have

$$\mathrm{E}[F(\mathbf{y}_j) - F(\mathbf{x}_j)] \le \frac{C\gamma^3}{2}$$

Before termination, NEON returns $\mathbf{u}_j \ne 0$, then it satisfies $\frac{\mathbf{u}_j^\top \nabla^2 F(\mathbf{y}_j) \mathbf{u}_j}{\|\mathbf{u}_j\|^2} \le -c\gamma$ with high probability according to Lemma 1 and Lemma 3, i.e., $\Pr\left( \frac{\mathbf{u}_j^\top \nabla^2 F(\mathbf{y}_j) \mathbf{u}_j}{\|\mathbf{u}_j\|^2} \le -c\gamma \right) \ge 1 - \delta$. Similar to the analysis in Lemma 1, we have

$$\mathrm{E}[F(\mathbf{y}_j + c\gamma \bar{\xi} \hat{\mathbf{u}}/L_2) - F(\mathbf{y}_j)] \le \mathrm{E}\left[ \frac{1}{2} \eta^2 \hat{\mathbf{u}}^\top \nabla^2 F(\mathbf{y}_j) \hat{\mathbf{u}} + \frac{L_2}{6} \|\eta \hat{\mathbf{u}}\|^3 \right],$$

where $\hat{\mathbf{u}} = \mathbf{u}_j/\|\mathbf{u}_j\|$ and $\eta = c\gamma\bar{\xi}/L_2$.

$$
\begin{aligned}
\mathrm{E}[\hat{\mathbf{u}}^\top \nabla^2 F(\mathbf{y}_j)\hat{\mathbf{u}}] =& \mathrm{E}[\hat{\mathbf{u}}^\top \nabla^2 F(\mathbf{y}_j)\hat{\mathbf{u}}|\hat{\mathbf{u}}^\top \nabla^2 F(\mathbf{y}_j)\hat{\mathbf{u}} \le -c\gamma] \cdot \Pr(\hat{\mathbf{u}}^\top \nabla^2 F(\mathbf{y}_j)\hat{\mathbf{u}} \le -c\gamma) \\
& + \mathrm{E}[\hat{\mathbf{u}}^\top \nabla^2 F(\mathbf{y}_j)\hat{\mathbf{u}}|\hat{\mathbf{u}}^\top \nabla^2 F(\mathbf{y}_j)\hat{\mathbf{u}} > -c\gamma] \cdot \Pr(\hat{\mathbf{u}}^\top \nabla^2 F(\mathbf{y}_j)\hat{\mathbf{u}} > -c\gamma) \\
\le& - c\gamma \Pr(\hat{\mathbf{u}}^\top \nabla^2 F(\mathbf{y}_j)\hat{\mathbf{u}} \le -c\gamma) + L_1\delta \\
\le& - c\gamma(1-\delta) + L_1\delta = -c\gamma + (c\gamma + L_1)\delta.
\end{aligned}
$$

With $\delta \le c\gamma/2(c\gamma + L_1)$, we have $\mathrm{E}[\hat{\mathbf{u}}^\top \nabla^2 F(\mathbf{y}_j)\hat{\mathbf{u}}] \le -c\gamma/2$. As a result,

$$
\mathrm{E}[F(\mathbf{x}_{j+1}) - F(\mathbf{y}_j)] \le \mathrm{E}\left[\frac{1}{2}\eta^2\hat{\mathbf{u}}^\top \nabla^2 F(\mathbf{y}_j)\hat{\mathbf{u}} + \frac{L_2}{6}\|\eta\hat{\mathbf{u}}\|^3\right] \le -\frac{c^3\gamma^3}{12L_2^2}
$$

By selecting $C$ such that $C < \frac{c^3}{6L_2^2}$, we will have

$$
\mathrm{E}[F(\mathbf{x}_{j+1}) - F(\mathbf{x}_j)] \le -C\gamma^3/2. \tag{15}
$$

By (14) and (15) we get

$$
\mathrm{E}[F(\mathbf{x}_{j+1}) - F(\mathbf{x}_j)] \le - \underbrace{\min(\varepsilon(\epsilon, \alpha), C\gamma^3/2)}_{\theta}. \tag{16}
$$

Next, we will show within $\widetilde{O}\left(\max\left(\frac{1}{\varepsilon(\epsilon,\alpha)}, \frac{1}{\gamma^3}\right)\log(1/\zeta)\right)$ outer iterations of NEON-$\mathcal{A}$, there exists at least on $\mathbf{y}_j$ such that $\lambda_{\min}(\nabla^2 F(\mathbf{y}_j)) \ge -\gamma$ and $\|\nabla F(\mathbf{y}_j)\| \le \epsilon$ with high probability. This analysis is similar to that of Theorem 14 in [5]. As a result, at such a $\mathbf{y}_j$ NEON-$\mathcal{A}$ terminates with a high probability. Let us consider three cases:

$$
\begin{aligned}
\mathcal{C}_1 =& \{\mathbf{y}_j | \|\nabla F(\mathbf{y}_j)\| \ge \epsilon \text{ for some } j > 0\} \\
\mathcal{C}_2 =& \{\mathbf{y}_j | \|\nabla F(\mathbf{y}_j)\| \le \epsilon \text{ and } \lambda_{\min}(\nabla^2 F(\mathbf{y}_j)) \le -\gamma \text{ for some } j > 0\} \\
\mathcal{C}_3 =& \{\mathbf{y}_j | \|\nabla F(\mathbf{y}_j)\| \le \epsilon \text{ and } \lambda_{\min}(\nabla^2 F(\mathbf{y}_j)) \ge -\gamma \text{ for some } j > 0\}
\end{aligned}
$$

Clearly, Case 3 is our favorable case, thus we need to carefully study the occurrences of Cases 1 and 2. Let us define an event $\mathcal{E}_j = \{\exists i \le j, \mathbf{y}_i \in \mathcal{C}_3\}$, and then $\bar{\mathcal{E}}_j = \{\forall i \le j, \mathbf{y}_i \notin \mathcal{C}_3\}$. It is easy to show that $Pr(\bar{\mathcal{E}}_j) \le Pr(\bar{\mathcal{E}}_{j-1})$. Then we have

$$
\mathrm{E}[F(\mathbf{x}_{j+1})I_{\bar{\mathcal{E}}_j}] - \mathrm{E}[F(\mathbf{x}_j)I_{\bar{\mathcal{E}}_{j-1}}] = \mathrm{E}[F(\mathbf{x}_{j+1}) - F(\mathbf{x}_j)|\bar{\mathcal{E}}_j]Pr(\bar{\mathcal{E}}_j) + \mathrm{E}[F(\mathbf{x}_j)I_{\bar{\mathcal{E}}_j}] - \mathrm{E}[F(\mathbf{x}_j)I_{\bar{\mathcal{E}}_{j-1}}]
$$

It is easy to bound the first term in R.H.S by $\mathrm{E}[F(\mathbf{x}_{j+1}) - F(\mathbf{x}_j)|\bar{\mathcal{E}}_j]Pr(\bar{\mathcal{E}}_j) \le -\theta Pr(\bar{\mathcal{E}}_j)$. To bound the second term, we need following two results.

(a) Let consider different situations of $I_{\bar{\mathcal{E}}_j}$ and $I_{\bar{\mathcal{E}}_{j-1}}$.

- If $I_{\bar{\mathcal{E}}_j} = 1$, then $I_{\bar{\mathcal{E}}_{j-1}} = 1$, so that
$$
\mathrm{E}[F(\mathbf{x}_j)I_{\bar{\mathcal{E}}_j}] - \mathrm{E}[F(\mathbf{x}_j)I_{\bar{\mathcal{E}}_{j-1}}] = \mathrm{E}[F(\mathbf{x}_j)] - \mathrm{E}[F(\mathbf{x}_j)] = 0.
$$

- If $I_{\bar{\mathcal{E}}_j} = 0$, then $I_{\bar{\mathcal{E}}_{j-1}}$ could be either 1 or 0. When $I_{\bar{\mathcal{E}}_{j-1}} = 0$, then
$$
\mathrm{E}[F(\mathbf{x}_j)I_{\bar{\mathcal{E}}_j}] - \mathrm{E}[F(\mathbf{x}_j)I_{\bar{\mathcal{E}}_{j-1}}] = 0.
$$
When $I_{\bar{\mathcal{E}}_{j-1}} = 1$, then
$$
\mathrm{E}[F(\mathbf{x}_j)I_{\bar{\mathcal{E}}_j}] - \mathrm{E}[F(\mathbf{x}_j)I_{\bar{\mathcal{E}}_{j-1}}] = -\mathrm{E}[F(\mathbf{x}_j)]\Pr(\bar{\mathcal{E}}_{j-1} - \bar{\mathcal{E}}_j).
$$
Please note that here $\Pr(\bar{\mathcal{E}}_{j-1} - \bar{\mathcal{E}}_j)$ means the probability of $\bar{\mathcal{E}}_{j-1}$ happens (i.e., $I_{\bar{\mathcal{E}}_{j-1}} = 1$) and $\bar{\mathcal{E}}_j$ doesn't happen (i.e., $I_{\bar{\mathcal{E}}_j} = 0$).

(b) According to (16) we can see that under the event $\bar{\mathcal{E}}_j$
$$
\mathrm{E}[F(\mathbf{x}_j)] \le F(\mathbf{x}_0) - j\theta.
$$
As a result, $F(\mathbf{x}_*) \le \mathrm{E}[F(\mathbf{x}_j)] \le F(\mathbf{x}_0)$, and
$$
|\mathrm{E}[F(\mathbf{x}_j)]| \le B := \max\{|F(\mathbf{x}_*)|, |F(\mathbf{x}_0)|\}. \tag{17}
$$
Thus,
$$
\mathrm{E}[F(\mathbf{x}_{j+1})I_{\bar{\mathcal{E}}_j}] - \mathrm{E}[F(\mathbf{x}_j)I_{\bar{\mathcal{E}}_{j-1}}] \le -\theta\Pr(\bar{\mathcal{E}}_j) + B\Pr(\bar{\mathcal{E}}_{j-1} - \bar{\mathcal{E}}_j).
$$

By summing up over $k = 0, \ldots, j$ we have

$$\mathrm{E}[F(\mathbf{x}_{j+1})I_{\bar{\mathcal{E}}_j}] - \mathrm{E}[F(\mathbf{x}_0)] \leq -\theta \sum_{k=0}^{j} \Pr(\bar{\mathcal{E}}_k) + B \sum_{k=0}^{j} \Pr(\bar{\mathcal{E}}_{k-1} - \bar{\mathcal{E}}_k)$$

$$\leq -\theta \sum_{k=0}^{j} \Pr(\bar{\mathcal{E}}_k) + B\Pr(-\bar{\mathcal{E}}_j) \leq -\theta(j+1)\Pr(\bar{\mathcal{E}}_j) + B\Pr(-\bar{\mathcal{E}}_j),$$

which implies

$$\theta(j+1)\Pr(\bar{\mathcal{E}}_j) \leq B\Pr(-\bar{\mathcal{E}}_j) - E[F(\mathbf{x}_{j+1})I_{\bar{\mathcal{E}}_j}] + E[F(\mathbf{x}_0)]$$
$$\leq B + [F(\mathbf{x}_0) - F(\mathbf{x}_*)] \leq B + \Delta.$$

As $(j+1)$ grows to as large as $\frac{2(B+\Delta)}{\theta}$, we will have $Pr(\bar{\mathcal{E}}_j) \leq \frac{1}{2}$. Therefore, after $\widetilde{O}(\frac{2(B+\Delta)}{\theta})$ steps, $\mathbf{y}_j \in \mathcal{C}_3$ must occur at least once with probability at least $\frac{1}{2}$. If we repeat this $\log(1/\zeta)$ times, then after $\widetilde{O}(\log(1/\zeta)\frac{1}{\theta})$ steps, with probability at least $1 - \zeta/2$, $\mathbf{y}_j \in \mathcal{C}_3$ must occur at least once. Therefore, with high probability, NEON-$\mathcal{A}$ terminates with a total IFO complexity of $\widetilde{O}(\max\left(\frac{1}{\varepsilon(\epsilon,\alpha)}, \frac{1}{\gamma^3}\right)(T_n + T_a + T_c))$. $\qquad\square$

# E   Proof of Corollary 5

*Proof.* Let us first consider $\mathrm{SM}(\mathbf{x}_0, \eta, \beta, s, t)$. According to the analysis of Theorem 3 in [23], when $\eta \leq (1-\beta)/(2L_1)$, we have

$$\frac{1}{t+1} \sum_{\tau=0}^{t} \mathrm{E}[\|\nabla F(\mathbf{x}_\tau)\|^2] \leq \frac{2\mathrm{E}[F(\mathbf{x}_0) - F(\mathbf{x}_{t+1}^+)]}{\eta(t+1)/(1-\beta)} + D\eta,$$

where $D = \left[\frac{L_1\beta^2((1-\beta)s-1)^2\sigma^2}{(1-\beta)^3} + \frac{L_1\sigma^2}{1-\beta}\right]$, and $\sigma^2$ is the upper bound of $\mathrm{E}[\|\nabla f(\mathbf{x};\xi)\|^2]$. On the other hand, by Lemma 3 of [23], we know for any $\tau \geq 0$,

$$\mathrm{E}[\|\nabla F(\mathbf{x}_\tau^+) - \nabla F(\mathbf{x}_\tau)\|^2] \leq \frac{L_1\eta^2}{1-\beta}\left[D - \frac{L_1\sigma^2}{1-\beta}\right] \leq \frac{L_1\eta^2}{1-\beta}D \leq \frac{D\eta}{2},$$

where the last inequality is due to $\eta \leq (1-\beta)/(2L_1)$. Since $\mathrm{E}[\|\nabla F(\mathbf{x}_\tau^+)\|^2] \leq 2\mathrm{E}[\|\nabla F(\mathbf{x}_\tau^+) - \nabla F(\mathbf{x}_\tau)\|^2 + \|\nabla F(\mathbf{x}_\tau)\|^2]$, then

$$\frac{1}{t+1} \sum_{\tau=0}^{t} \mathrm{E}[\|\nabla F(\mathbf{x}_\tau^+)\|^2] \leq \frac{4\mathrm{E}[F(\mathbf{x}_0) - F(\mathbf{x}_{t+1}^+)]}{\eta(t+1)/(1-\beta)} + 3D\eta, \tag{18}$$

Next, consider $(\mathbf{y}_j, \mathbf{z}_j) = \mathrm{SM}(\mathbf{x}_j, \eta, \beta, s, t)$, we have $\mathbf{y}_j = \mathbf{x}_{\tau'}^+$, $\mathbf{z}_j = \mathbf{x}_{t+1}^+$, and

$$\mathrm{E}[\|\nabla F(\mathbf{y}_j)\|^2] \leq \frac{4\mathrm{E}[F(\mathbf{x}_j) - F(\mathbf{z}_j)]}{\eta(t+1)/(1-\beta)} + 3D\eta.$$

When $\|\nabla F(\mathbf{y}_j)\| \geq \epsilon$ we have

$$\mathrm{E}[F(\mathbf{z}_j) - F(\mathbf{x}_j)] \leq -\frac{\eta(t+1)}{4(1-\beta)}(\epsilon^2 - 3D\eta) \leq -\frac{C'\epsilon^2}{48D(1-\beta)},$$

where the last inequality holds by choosing $\eta = \frac{\epsilon^2}{6D}$ and $t+1 \geq \frac{C'}{\epsilon^2}$, where $C'$ is a constant. Therefore, we have $\varepsilon(\epsilon,\alpha) = \frac{C'\epsilon^2}{48D(1-\beta)}$. Similarly, from (18) we have

$$\mathrm{E}[F(\mathbf{y}_j) - F(\mathbf{x}_j)] \leq \frac{3\eta^2(t+1)D}{4(1-\beta)} \leq \frac{C\epsilon^2}{2},$$

where the last inequality holds with $t+1 \leq \frac{24DC(1-\beta)}{\epsilon^2}$. Therefore, when $\|\nabla F(\mathbf{y}_j)\| \leq \epsilon$, we have $\mathrm{E}[F(\mathbf{y}_j) - F(\mathbf{x}_j)] \leq \epsilon^2$. By assuming that $\gamma \geq \epsilon^{2/3}$, the two inequalities in (9) hold. $\qquad\square$

# F  Proof of Corollary 6

*Proof.* Based on Theorem 4, we only need to show (9) holds for NEON-MSGD. By the updates in (12), we know $\mathbf{y}_j = \mathbf{x}_j$. It implies that the second inequality of (9) holds, i.e. $\mathrm{E}[F(\mathbf{y}_j) - F(\mathbf{x}_j)] \leq \frac{C\gamma^3}{2}$. Next, we will show the first inequalty holds when $\|\nabla F(\mathbf{y}_j)\| \geq \epsilon$, i.e. $\|\nabla F(\mathbf{x}_j)\| \geq \epsilon$. Recall that the update of MSGD is $\mathbf{z}_j = \mathbf{x}_j - \frac{1}{L_1}\nabla F_{\mathcal{S}_1}(\mathbf{x}_j)$, then by the smoothness of $F(\cdot)$ we have

$$
\begin{aligned}
\mathrm{E}[F(\mathbf{z}_j) - F(\mathbf{x}_j)] \leq & \mathrm{E}\left[(\mathbf{z}_j - \mathbf{x}_j)^\top \nabla F(\mathbf{x}_j) + \frac{L_1}{2}\|\mathbf{z}_j - \mathbf{x}_j\|^2\right] \\
= & -\frac{1}{L_1}\|\nabla F(\mathbf{x}_j)\|^2 + \mathrm{E}\left[\frac{1}{2L_1}\|\nabla F_{\mathcal{S}_1}(\mathbf{x}_j)\|^2\right] \\
= & -\frac{1}{2L_1}\|\nabla F(\mathbf{x}_j)\|^2 + \mathrm{E}\left[\frac{1}{2L_1}\|\nabla F(\mathbf{x}_j) - \nabla F_{\mathcal{S}_1}(\mathbf{x}_j)\|^2\right] \\
\leq & -\frac{1}{2L_1}\|\nabla F(\mathbf{x}_j)\|^2 + \frac{1}{2L_1}\frac{V}{|\mathcal{S}_1|},
\end{aligned}
$$

where the last inequality uses Assumption 1 (iv). By choosing $|\mathcal{S}_1| \geq \frac{2V}{\epsilon^2}$ and using the condition of $\|\nabla F(\mathbf{x}_j)\| \geq \epsilon$, we have

$$
\mathrm{E}[F(\mathbf{z}_j) - F(\mathbf{x}_j)] \leq -\frac{\epsilon^2}{4L_1} = -\varepsilon(\epsilon, \alpha).
$$

Additionally, we have $\mathbf{y}_j = \mathbf{x}_j$ indicating that the second inequality in (9) holds. ◻

# G  Proof of Corollary 7

*Proof.* Based on Theorem 4, we only need to show (9) holds for NEON-SCSG. By the updates in (13), we know $\mathbf{z}_j = \mathbf{y}_j$. Each epoch of SCGS guarantees [14] that

$$
\mathrm{E}[\|\nabla F(\mathbf{y}_j)\|^2] \leq \frac{5L_1 b^{1/3}}{c'|\mathcal{S}_1|^{1/3}}\mathrm{E}[F(\mathbf{x}_j) - F(\mathbf{y}_j)] + \frac{6V}{|\mathcal{S}_1|}.
$$

As a result, when $\|\nabla F(\mathbf{y}_j)\| \geq \epsilon$, we have

$$
\mathrm{E}[F(\mathbf{y}_j) - F(\mathbf{x}_j)] \leq -\frac{c'|\mathcal{S}_1|^{1/3}}{5L_1 b^{1/3}}\left(\epsilon^2 - \frac{6V}{|\mathcal{S}_1|}\right)
$$

By setting $|\mathcal{S}_1| \geq 12V/\epsilon^2$, we have

$$
\mathrm{E}[F(\mathbf{y}_j) - F(\mathbf{x}_j)] \leq -\frac{c'(12V)^{1/3}}{10L_1}\frac{\epsilon^{4/3}}{b^{1/3}}
$$

On the other hand, when $\|\nabla F(\mathbf{y}_j)\| \leq \epsilon$, we have

$$
0 \leq \mathrm{E}[\|\nabla F(\mathbf{y}_j)\|^2] \leq \frac{5L_1 b^{1/3}}{c'|\mathcal{S}_1|^{1/3}}\mathrm{E}[F(\mathbf{x}_j) - F(\mathbf{y}_j)] + \frac{6V}{|\mathcal{S}_1|},
$$

i.e.,

$$
\mathrm{E}[F(\mathbf{y}_j) - F(\mathbf{x}_j)] \leq \frac{6c'V}{5L_1|\mathcal{S}_1|^{2/3}b^{1/3}}.
$$

By choosing $|\mathcal{S}_1| \geq \left(\frac{12Vc'}{5CL_1}\right)^{3/2}\frac{1}{b^{1/2}\gamma^{9/2}}$,

$$
\mathrm{E}[F(\mathbf{y}_j) - F(\mathbf{x}_j)] \leq \frac{C\gamma^3}{2}.
$$

In summary, the sample sizes must satisfy $|\mathcal{S}_1| \geq \widetilde{O}(\max(1/\epsilon^2, 1/(\gamma^{9/2}b^{1/2})))$. The total iterations (i.e., the number of calls of SCSG-Epoch and NEON) is $\widetilde{O}\left(\max(\frac{b^{1/3}}{\epsilon^{4/3}}, \frac{1}{\gamma^3})\right)$. The expected IFO complexity of each SCSG-Epoch is $2|\mathcal{S}_1| \geq \widetilde{O}(\max(1/\epsilon^2, 1/(\gamma^{9/2}b^{1/2})))$ regardless the value of $b$ due to the geometric distribution of $N$ in SCSG-Epoch. ◻

# H   Proof of Theorem 1

We first prove the following lemma.

**Lemma 4.** *Under the same conditions and settings as in Theorem 1, If NEON is called at a point $\mathbf{x}$ such that $\lambda_{\min}(\nabla^2 f(\mathbf{x})) \leq -\gamma$, then with high probability $1 - \delta$ there exists $\tau \leq t$ such that*

$$f(\mathbf{x} + \mathbf{u}_\tau) - f(\mathbf{x}) - \nabla f(\mathbf{x})^\top \mathbf{u}_\tau \leq -2\mathcal{F},$$
$$\|\mathbf{u}_\tau\| \leq U = 4\hat{c}(\sqrt{\eta L_1}\mathcal{F}/L_2)^{1/3}$$

For simplicity, we recall and define some notations. Let $H = \nabla^2 f(\mathbf{x})$ be the Hessian matrix and $\mathbf{e}_1$ be the unit minimum eigenvector of $H$. We also use the following notations in the proofs:

$$\mathcal{F} := \eta L_1 \frac{\gamma^3}{L_2^2} \cdot \log^{-3}(d\kappa/\delta),$$

$$\mathcal{P} := \sqrt{\eta L_1} \frac{\gamma}{L_2} \cdot \log^{-1}(d\kappa/\delta),$$

$$\mathcal{J} := \frac{\log(d\kappa/\delta)}{\eta\gamma},$$

where $\kappa = L_1/\gamma \geq 1$ is the condition number. From above notations, we know $r = \frac{\mathcal{P}}{\kappa}\log^{-1}(d\kappa/\delta) = \mathcal{P}\gamma L_1^{-1}\log^{-1}(d\kappa/\delta) \leq \mathcal{P}$. Let us define

$$\hat{f}_\mathbf{x}(\mathbf{u}) = f(\mathbf{x} + \mathbf{u}) - f(\mathbf{x}) - \mathbf{u}^\top \nabla f(\mathbf{x}) \tag{19}$$

Also, we need the following two lemmas for our proof. It is notable that the following two lemmas are similar to Lemma 16 and Lemma 17 in [11]. The analysis of Lemma 6 is similar to that of Lemma 17 in [11]. However, we provide a much simpler analysis for Lemma 5 than that of Lemma 16 in [11].

**Lemma 5.** *For any constant $\hat{c} \geq 8$, there exists $c_{\max}$: for $\mathbf{x}$ satisfies the condition that $\lambda_{\min}(\nabla^2 f(\mathbf{x})) \leq -\gamma$ and any initial point $\mathbf{u}_0$ with $\|\mathbf{u}_0\| \leq 2r$, define*

$$T = \min\{\inf_\tau\{\tau|\hat{f}_\mathbf{x}(\mathbf{u}_\tau) - \hat{f}_\mathbf{x}(\mathbf{u}_0) \leq -3\mathcal{F}\}, \hat{c}\mathcal{J}\}.$$

*Then for any $\eta \leq c_{\max}/L_1$, we have for all $\tau < T$ that $\|\mathbf{u}_\tau\| \leq 2\hat{c}\mathcal{P}$.*

**Lemma 6.** *There exist constant $c_{\max}, \hat{c}$ such that: for $\mathbf{x}$ satisfies the condition that $\lambda_{\min}(\nabla^2 f(\mathbf{x})) \leq -\gamma$ define a sequence $\mathbf{w}_t$ similar to $\mathbf{u}_t$ except $\mathbf{w}_0 = \mathbf{u}_0 + \mu r \mathbf{e}_1$, where $\mu \in [\delta/2\sqrt{d}, 1]$ and $\mathbf{e}_1$ is a unit eigen-vector corresponding to the minimum eigen-value of $\nabla^2 f(\mathbf{x})$, and let $\mathbf{v}_t = \mathbf{w}_t - \mathbf{u}_t$*

$$T = \min\{\inf_\tau\{\tau|\hat{f}_\mathbf{x}(\mathbf{w}_\tau) - \hat{f}_\mathbf{x}(\mathbf{w}_0) \leq -3\mathcal{F}\}, \hat{c}\mathcal{J}\}.$$

*Then for any $\eta \leq c_{\max}/L_1$, if $\|\mathbf{u}_\tau\| \leq 2\hat{c}\mathcal{P}$ for all $\tau < T$, then $T < \hat{c}\mathcal{J}$.*

*Proof.* **(of Lemma 4)** We define $T_* = \hat{c}\mathcal{J}$ and $T' = \inf_\tau\{\tau|\hat{f}_\mathbf{x}(\mathbf{u}_\tau) - \hat{f}_\mathbf{x}(\mathbf{u}_0) \leq -3\mathcal{F}\}$. Let's consider following two scenarios:

**(1)** $T' \leq T_*$: Since $\mathbf{u}_{T'} = \mathbf{u}_{T'-1} - \eta(\nabla f(\mathbf{x} + \mathbf{u}_{T'-1}) - \nabla f(\mathbf{x}))$, we can see that $\|\mathbf{u}_{T'}\| \leq \|\mathbf{u}_{T'-1}\| + \eta L_1 \|\mathbf{u}_{T'-1}\| \leq 4\hat{c}\mathcal{P} \triangleq U$ by employing Lemma 5. Then we have

$$f(\mathbf{x} + \mathbf{u}_{T'}) - f(\mathbf{x}) - \mathbf{u}_{T'}^\top \nabla f(\mathbf{x}) \leq f(\mathbf{x} + \mathbf{u}_0) - f(\mathbf{x}) - \mathbf{u}_0^\top \nabla f(\mathbf{x}) - 3\mathcal{F}$$
$$\leq \frac{L_1}{2}\|\mathbf{u}_0\|^2 - 3\mathcal{F} \leq \mathcal{F} - 3\mathcal{F} = -2\mathcal{F}.$$

Therefore we have

$$\min_{1\leq\tau'\leq T_*, \|\mathbf{u}_{\tau'}\|\leq U} f(\mathbf{x} + \mathbf{u}_{\tau'}) - f(\mathbf{x}) - \mathbf{u}_{\tau'}^\top \nabla f(\mathbf{x}) \leq f(\mathbf{x} + \mathbf{u}_{T'}) - f(\mathbf{x}) - \mathbf{u}_{T'}^\top \nabla f(\mathbf{x}) \leq -2\mathcal{F}.$$

**(2)** $T' > T_*$: By Lemma 5, we have $\|\mathbf{u}_\tau\| \leq 2\hat{c}\mathcal{P}$ for all $\tau \leq T_*$. Let define $T'' = \inf_\tau\{\tau|\hat{f}_\mathbf{x}(\mathbf{w}_\tau) - \hat{f}_\mathbf{x}(\mathbf{w}_0) \leq -3\mathcal{F}\}$. By Lemma 6, we konw $T'' \leq T_*$. From the proof of Lemma 6, we also know that $\|\mathbf{w}_{T''-1}\| \leq 2\hat{c}\mathcal{P}$. Hence $\|\mathbf{w}_{T''}\| \leq \|\mathbf{w}_{T''-1}\| + \eta L_1 \|\mathbf{w}_{T''-1}\| \leq 4\hat{c}\mathcal{P}$. Similar to case (1), we have

$$\min_{1\leq\tau'\leq T_*, \|\mathbf{w}_{\tau'}\|\leq U} f(\mathbf{x} + \mathbf{w}_{\tau'}) - f(\mathbf{x}) - \mathbf{w}_{\tau'}^\top \nabla f(\mathbf{x}) \leq -2\mathcal{F}.$$

Therefore,

$$\min \left\{ \min_{1 \leq \tau' \leq T_*, \|\mathbf{u}_{\tau'}\| \leq U} f(\mathbf{x} + \mathbf{u}_{\tau'}) - f(\mathbf{x}) - \mathbf{u}_{\tau'}^\top \nabla f(\mathbf{x}), \right.$$
$$\left. \min_{1 \leq \tau' \leq T_*, \|\mathbf{w}_{\tau'}\| \leq U} f(\mathbf{x} + \mathbf{w}_{\tau'}) - f(\mathbf{x}) - \mathbf{w}_{\tau'}^\top \nabla f(\mathbf{x}) \right\} \leq -2\mathcal{F}.$$

We know $\mathbf{u}_0$ follows an uniform distribution over $\mathbb{B}_0(r)$ with radius $r = \mathcal{P}/(\kappa \cdot \log(d\kappa/\delta))$. Let denote by $\mathcal{X}_s \subset \mathbb{B}_0(r)$ the set of bad initial points such that $\min_{1 \leq \tau' \leq T_*, \|\mathbf{u}'_\tau\| \leq U} f(\mathbf{x} + \mathbf{u}_{\tau'}) - f(\mathbf{x}) - \mathbf{u}_{\tau'}^\top \nabla f(\mathbf{x}) > -2\mathcal{F}$ when $\mathbf{u}_0 \in \mathcal{X}_s$; otherwise $\min_{1 \leq \tau' \leq T_*, \|\mathbf{u}'_\tau\| \leq U} f(\mathbf{x} + \mathbf{u}_{\tau'}) - f(\mathbf{x}) - \mathbf{u}_{\tau'}^\top \nabla f(\mathbf{x}) \leq -2\mathcal{F}$ when $\mathbf{u}_0 \in \mathbb{B}_0(r) - \mathcal{X}_s$.

By above analysis, for any $\mathbf{u}_0 \in \mathcal{X}_s$, we have $(\mathbf{u}_0 \pm \mu r \mathbf{e}_1) \notin \mathcal{X}_s$ where $\mu \in [\frac{\delta}{2\sqrt{d}}, 1]$. Let denote by $I_{\mathcal{X}_s}(\cdot)$ the indicator function of being inside set $\mathcal{X}_s$. We set $u^{(1)}$ as the component along $\mathbf{e}_1$ direction and $\mathbf{u}^{(-1)}$ as the remaining $d - 1$ dimensional vector, then the vector $\mathbf{u} = (u^{(1)}, \mathbf{u}^{(-1)})$. It is easy to have an upper bound of $\mathcal{X}_s$'s volumn:

$$\text{Vol}(\mathcal{X}_s) = \int_{\mathbb{B}_0^{(d)}(r)} d\mathbf{u} \cdot I_{\mathcal{X}_s}(\mathbf{u}) = \int_{\mathbb{B}_0^{(d-1)}(r)} d\mathbf{u}^{(-1)} \int_{-\sqrt{r^2 - \|\mathbf{u}^{(-1)}\|^2}}^{\sqrt{r^2 - \|\mathbf{u}^{(-1)}\|^2}} du^{(1)} \cdot I_{\mathcal{X}_s}(\mathbf{u})$$
$$\leq \int_{\mathbb{B}_0^{(d-1)}(r)} d\mathbf{u}^{(-1)} \cdot 2\mu r \leq \int_{\mathbb{B}_0^{(d-1)}(r)} d\mathbf{u}^{(-1)} \cdot 2 \frac{\delta}{2\sqrt{d}} r = \text{Vol}(\mathbb{B}_0^{(d-1)}(r)) \frac{\delta r}{\sqrt{d}}$$

Then,

$$\frac{\text{Vol}(\mathcal{X}_s)}{\text{Vol}(\mathbb{B}_0^{(d)}(r))} \leq \frac{\text{Vol}(\mathbb{B}_0^{(d-1)}(r)) \frac{\delta r}{\sqrt{d}}}{\text{Vol}(\mathbb{B}_0^{(d)}(r))} = \frac{\delta}{\sqrt{\pi d}} \frac{\Gamma(\frac{d}{2} + 1)}{\Gamma(\frac{d}{2} + \frac{1}{2})} \leq \frac{\delta}{\sqrt{\pi d}} \cdot \sqrt{\frac{d}{2} + \frac{1}{2}} \leq \delta$$

where the second inequality is due to $\frac{\Gamma(x+1)}{\Gamma(x+1/2)} < \sqrt{x + \frac{1}{2}}$ for all $x \geq 0$. Thus, we have $\mathbf{u}_0 \notin \mathcal{X}_s$ with at least probability $1 - \delta$. Therefore,

$$\min_{1 \leq \tau' \leq T_*, \|\mathbf{u}_{\tau'}\| \leq U} f(\mathbf{x} + \mathbf{u}_{\tau'}) - f(\mathbf{x}) - \mathbf{u}_{\tau'}^\top \nabla f(\mathbf{x}) \leq -2\mathcal{F}. \tag{20}$$

To finish the proof of Theorem 1, by the Lipschitz continuity of Hessian, we have

$$\left| f(\mathbf{x} + \mathbf{u}_\tau) - f(\mathbf{x}) - \nabla f(\mathbf{x})^\top \mathbf{u}_\tau - \frac{1}{2} \mathbf{u}_\tau^\top \nabla^2 f(\mathbf{x}) \mathbf{u}_\tau \right| \leq \frac{L_2}{6} \|\mathbf{u}_\tau\|^3.$$

Then by inequality (20), we get

$$\frac{1}{2} \mathbf{u}_\tau^\top \nabla^2 f(\mathbf{x}) \mathbf{u}_\tau \leq f(\mathbf{x} + \mathbf{u}_\tau) - f(\mathbf{x}) - \nabla f(\mathbf{x})^\top \mathbf{u}_\tau + \frac{L_2}{6} \|\mathbf{u}_\tau\|^3 \leq -2\mathcal{F} + \mathcal{F} \leq -\mathcal{F}.$$

That is,

$$\frac{\mathbf{u}_\tau^\top \nabla^2 f(\mathbf{x}) \mathbf{u}_\tau}{\|\mathbf{u}_\tau\|^2} \leq \frac{-2\mathcal{F}}{(4\hat{c}\mathcal{P})^2} = -\frac{\gamma}{8\hat{c}^2 \log(dL_1/(\gamma\delta))}. \tag{21}$$

If NEON returns 0 following Bayes theorem, it is not difficult to show that $\lambda_{\min}(\nabla^2 f(\mathbf{x})) \geq -\gamma$ with high probability $1 - O(\delta)$ for a sufficiently small $\delta$. $\qquad \square$

## H.1  Proof of Lemma 5

*Proof.* By using the smoothness of $\hat{f}_{\mathbf{x}}(\mathbf{u})$, we have

$$\hat{f}(\mathbf{u}_{\tau+1}) \leq \hat{f}(\mathbf{u}_\tau) + \nabla \hat{f}(\mathbf{u}_\tau)^\top (\mathbf{u}_{\tau+1} - \mathbf{u}_\tau) + \frac{L_1}{2} \|\mathbf{u}_{\tau+1} - \mathbf{u}_\tau\|^2$$
$$= \hat{f}(\mathbf{u}_\tau) - \frac{1}{\eta} \|\mathbf{u}_{\tau+1} - \mathbf{u}_\tau\|^2 + \frac{L_1}{2} \|\mathbf{u}_{\tau+1} - \mathbf{u}_\tau\|^2$$
$$\leq \hat{f}(\mathbf{u}_\tau) - \frac{1}{2\eta} \|\mathbf{u}_{\tau+1} - \mathbf{u}_\tau\|^2,$$

where the first equality uses the update of $\mathbf{u}_{\tau+1}$ in Algorithm 1; the last inequality uses the fact of $\eta L_1 < 1$. By summing up $\tau$ from $0$ to $t-1$ where $t < T$, we have

$$\hat{f}(\mathbf{u}_t) \leq \hat{f}(\mathbf{u}_0) - \frac{1}{2\eta} \sum_{\tau=0}^{t-1} \|\mathbf{u}_\tau - \mathbf{u}_{\tau-1}\|^2,$$

which inplies

$$\frac{1}{2\eta} \sum_{\tau=0}^{t-1} \|\mathbf{u}_\tau - \mathbf{u}_{\tau-1}\|^2 \leq \hat{f}(\mathbf{u}_0) - \hat{f}(\mathbf{u}_t).$$

Hence, if $t \leq T$ then $\hat{f}(\mathbf{u}_t) - \hat{f}(\mathbf{u}_0) \geq -3\mathcal{F}$, i.e., $\hat{f}(\mathbf{u}_0) - \hat{f}(\mathbf{u}_t) \leq 3\mathcal{F}$. Then we have

$$\sum_{\tau=0}^{t-1} \|\mathbf{u}_\tau - \mathbf{u}_{\tau-1}\| \leq \sqrt{t \sum_{\tau=0}^{t-1} \|\mathbf{u}_\tau - \mathbf{u}_{\tau-1}\|^2} \leq \sqrt{\frac{3\hat{c} \log(\frac{d\kappa}{\delta})}{\eta\gamma} \frac{\eta L_1 \gamma^3}{L_2^2 \log^3(\frac{d\kappa}{\delta})} 2\eta}$$

$$= \sqrt{6\hat{c}} \sqrt{\eta L_1} \frac{\gamma}{L_2} \log^{-1}(d\kappa/\delta) = \sqrt{6\hat{c}}\mathcal{P}.$$

Then for all $\tau \leq t-1$, $\|\mathbf{u}_\tau\| \leq \sum_{k=1}^{\tau} \|\mathbf{u}_k - \mathbf{u}_{k-1}\| + \|\mathbf{u}_0\| \leq \sqrt{6\hat{c}}\mathcal{P} + \mathcal{P} \leq \hat{c}\mathcal{P}$, where the last inequality is due to $\hat{c} \geq 18$. Additionally, by the update of $\mathbf{u}_{\tau+1}$ in Algorithm 1, we have

$$\|\mathbf{u}_{\tau+1}\| \leq (1 + \eta L_1) \|\mathbf{u}_\tau\| \leq 2\hat{c}\mathcal{P}.$$

$\square$

## H.2 Proof of Lemma 6

The proof is almost the same as the proof of Lemma 17 in [11]. For completeness, we include it in this subsection. By the way $\mathbf{u}_0$ is constructed, we have $\|\mathbf{u}_0\| \leq r \leq \hat{c}\mathcal{P}$. Let us define $\mathbf{v}_\tau = \mathbf{w}_\tau - \mathbf{u}_\tau$, then $\mathbf{v}_0 = \mu r \mathbf{e}_1$ (i.e., $r = \frac{\mathcal{P}}{\kappa} \log^{-1}(d\kappa/\delta)$), $\mu \in [\delta/(2\sqrt{d}), 1]$. By the update equation of $\mathbf{w}_{\tau+1}$ (similar to $\mathbf{u}_{\tau+1}$), we have

$$\mathbf{u}_{\tau+1} + \mathbf{v}_{\tau+1} = \mathbf{w}_{\tau+1} = \mathbf{w}_\tau - \eta(\nabla f(\mathbf{x} + \mathbf{w}_\tau) - \nabla f(\mathbf{x}))$$

$$= \mathbf{u}_\tau + \mathbf{v}_\tau - \eta(\nabla f(\mathbf{x} + \mathbf{u}_\tau + \mathbf{v}_\tau) - \nabla f(\mathbf{x}))$$

$$= \mathbf{u}_\tau + \mathbf{v}_\tau - \eta(\nabla f(\mathbf{x} + \mathbf{u}_\tau + \mathbf{v}_\tau) - \nabla f(\mathbf{x} + \mathbf{u}_\tau) + \nabla f(\mathbf{x} + \mathbf{u}_\tau) - \nabla f(\mathbf{x}))$$

$$= \mathbf{u}_\tau - \eta(\nabla f(\mathbf{x} + \mathbf{u}_\tau) - \nabla f(\mathbf{x})) + \mathbf{v}_\tau - \eta H \mathbf{v}_\tau - \eta \left[ \int_0^1 \nabla^2 f(\mathbf{x} + \mathbf{u}_\tau + \theta\mathbf{v}_\tau) \mathrm{d}\theta - H \right] \mathbf{v}_\tau$$

$$= \mathbf{u}_\tau - \eta(\nabla f(\mathbf{x} + \mathbf{u}_\tau) - \nabla f(\mathbf{x})) + [I - \eta H - \eta \Delta'_\tau] \mathbf{v}_\tau$$

where $\Delta'_\tau := \int_0^1 \nabla^2 f(\mathbf{x} + \mathbf{u}_\tau + \theta\mathbf{v}_\tau) \mathrm{d}\theta - H$. We know $\|\Delta'_\tau\| \leq L_2(\|\mathbf{u}_\tau\| + \|\mathbf{v}_\tau\|/2)$ according to the Lipschitz continuity of Hessian. Then we know the update of $\mathbf{v}_\tau$ is

$$\mathbf{v}_{\tau+1} = (I - \eta H - \eta \Delta'_\tau) \mathbf{v}_\tau \tag{22}$$

We first show that $\mathbf{v}_\tau, \tau < T$ is upper bounded. Due to $\|\mathbf{w}_0\| \leq \|\mathbf{u}_0\| + \|\mathbf{v}_0\| \leq r + r = 2r$, following the result in Lemma 5, we have $\|\mathbf{w}_\tau\| \leq 2\hat{c}\mathcal{P}$ for all $\tau < T$. According to the condition in Lemma 6, we have $\|\mathbf{u}_\tau\| \leq 2\hat{c}\mathcal{P}$ for all $\tau < T$. Thus

$$\|\mathbf{v}_\tau\| \leq \|\mathbf{u}_\tau\| + \|\mathbf{w}_\tau\| \leq 4\hat{c}\mathcal{P} \text{ for all } \tau < T \tag{23}$$

Then we have for $\tau < T$

$$\|\Delta'_\tau\| \leq L_2(\|\mathbf{u}_\tau\| + \|\mathbf{v}_\tau\|/2) \leq L_2 \cdot 2\hat{c}\mathcal{P}$$

Next, we show that $\mathbf{v}_\tau, \tau < T$ is lower bounded. To proceed the proof, denote by $\psi_\tau$ the norm of $\mathbf{v}_\tau$ projected onto $\mathbf{e}_1$ direction and denote by $\varphi_\tau$ the norm of $\mathbf{v}_\tau$ projected onto the remaining subspace. The update equation of $\mathbf{v}_{\tau+1}$ (22) implies

$$\psi_{\tau+1} \geq (1 + \gamma\eta)\psi_\tau - \zeta\sqrt{\psi_\tau^2 + \varphi_\tau^2},$$

$$\varphi_{\tau+1} \leq (1 + \gamma\eta)\varphi_\tau + \zeta\sqrt{\psi_\tau^2 + \varphi_\tau^2},$$

where $\zeta = \eta L_2 \mathcal{P}(4\hat{c})$. We then prove the following inequality holds for all $\tau < T$ by induction,

$$\varphi_\tau \leq 4\zeta\tau \cdot \psi_\tau \tag{24}$$

According to the definition of $\mathbf{v}_0$, we have $\varphi_0 = 0$, indicating that the inequality (24) holds for $\tau = 0$. Assume the inequality (24) holds for all $\tau \leq t$. We need to show that the inequality (24) holds for $t + 1 \leq T$. It is easy to have the following inequalities

$$4\zeta(t+1)\psi_{t+1} \geq 4\zeta(t+1)\left((1+\gamma\eta)\psi_t - \zeta\sqrt{\psi_t^2 + \varphi_t^2}\right),$$

$$\varphi_{t+1} \leq 4\zeta t(1+\gamma\eta)\psi_t + \zeta\sqrt{\psi_t^2 + \varphi_t^2}.$$

Then we only need to show

$$(1 + 4\zeta(t+1))\sqrt{\psi_t^2 + \varphi_t^2} \leq 4(1+\gamma\eta)\psi_t.$$

If we choose $\sqrt{c_{\max}} \leq \frac{1}{(4\hat{c})^2}$, and $\eta \leq c_{\max}/L_1$, then

$$4\zeta(t+1) \leq 4\zeta T \leq 4\eta L_2 \mathcal{P}(4\hat{c}) \cdot \hat{c}\mathcal{J} = 4\sqrt{\eta L_1}(4\hat{c})\hat{c} \leq 1.$$

This implies

$$4(1+\gamma\eta)\psi_t \geq 4\psi_t \geq 2\sqrt{2\psi_t^2} \geq (1 + 4\zeta(t+1))\sqrt{\psi_t^2 + \varphi_t^2}.$$

We then finish the induction. According to (24), we get $\varphi_\tau \leq 4\zeta\tau \cdot \psi_\tau \leq \psi_\tau$ so that

$$\psi_{\tau+1} \geq (1+\gamma\eta)\psi_\tau - \sqrt{2}\zeta\psi_\tau \geq (1 + \frac{\gamma\eta}{2})\psi_\tau, \tag{25}$$

where the last inequality is due to $\zeta = \eta L_2 \mathcal{P}(4\hat{c}) \leq \sqrt{c_{\max}}(4\hat{c})\gamma\eta \cdot \log^{-1}(d\kappa/\delta) < \frac{\gamma\eta}{4\hat{c}} < \frac{\gamma\eta}{2\sqrt{2}}$ with $\eta \leq c_{\max}/L_1$, $\log(d\kappa/\delta) \geq 1$, and $\hat{c} > 1$.

By the inequalities (23) and (25), for all $\tau < T$, we get

$$4(\mathcal{P} \cdot \hat{c}) \geq \|\mathbf{v}_\tau\| \geq \psi_\tau \geq (1 + \frac{\gamma\eta}{2})^\tau \psi_0 = (1 + \frac{\gamma\eta}{2})^\tau \mu r$$

$$= (1 + \frac{\gamma\eta}{2})^\tau \mu \frac{\mathcal{P}}{\kappa} \log^{-1}(d\kappa/\delta) \geq (1 + \frac{\gamma\eta}{2})^\tau \frac{\delta}{2\sqrt{d}} \frac{\mathcal{P}}{\kappa} \log^{-1}(d\kappa/\delta).$$

Then

$$T < \frac{\log(8\frac{\kappa\sqrt{d}}{\delta} \cdot \hat{c}\log(d\kappa/\delta))}{\log(1 + \frac{\gamma\eta}{2})} \leq \frac{5}{2}\frac{\log(8\frac{\kappa\sqrt{d}}{\delta} \cdot \hat{c}\log(d\kappa/\delta))}{\gamma\eta} \leq \frac{5}{2}(2 + \log(8\hat{c}))\mathcal{J},$$

where the last inequality uses the facts that $\delta \in (0, \frac{d\kappa}{e}]$ and $\log(d\kappa/\delta) \geq 1$. By choosing a large enough constant $\hat{c}$ satisfying $\frac{5}{2}(2 + \log(8\hat{c})) \leq \hat{c}$ ($\hat{c} \geq 18$ works), we have $T < \hat{c}\mathcal{J}$ and complete the proof.

# I Proof of Theorem 2

We first consider the case when NCFind is executed, i.e., there exists $\tau > 0$

$$\hat{f}(\mathbf{y}_\tau) < \hat{f}(\mathbf{u}_\tau) + \nabla\hat{f}(\mathbf{u}_\tau)^\top(\mathbf{y}_\tau - \mathbf{u}_\tau) - \frac{\gamma}{2}\|\mathbf{y}_\tau - \mathbf{u}_\tau\|^2, \tag{26}$$

and NCfind returns a non-zero vector $\mathbf{v}$. We will prove that $\mathbf{v}^\top \nabla^2 f(\mathbf{x})\mathbf{v}/\|\mathbf{v}\|_2^2 \leq -\widetilde{\Omega}(\gamma)$. There are two scenarios we need to consider.

**Scenario (1)** there exists $j \leq \tau$ such that

$$j = \arg\min_{0 \leq k \leq \tau}\{\|\mathbf{y}_k - \mathbf{u}_k\| \geq \zeta\sqrt{6\eta\mathcal{F}}\}.$$

Since $\mathbf{u}_k - \mathbf{y}_k = \zeta(\mathbf{y}_k - \mathbf{y}_{k-1})$, then it is equivalent to

$$j = \arg\min_{0 \leq k \leq \tau}\{\|\mathbf{y}_k - \mathbf{y}_{k-1}\| \geq \sqrt{6\eta\mathcal{F}}\}.$$

According to the definition of $j$, we know for all $0 \leq k \leq j - 1$, $\|\mathbf{y}_k - \mathbf{y}_{k-1}\| \leq \sqrt{6\eta\mathcal{F}}$ and $\|\mathbf{y}_j - \mathbf{y}_{j-1}\| \geq \sqrt{6\eta\mathcal{F}}$. Thus,

$$\|\mathbf{y}_{j-1}\| \leq \sum_{k=1}^{j-1}\|\mathbf{y}_k - \mathbf{y}_{k-1}\| + \|\mathbf{y}_0\| \leq j\sqrt{6\eta\mathcal{F}} + 2\mathcal{P} \leq \hat{c}\mathcal{J}\sqrt{6\eta\mathcal{F}} + 2\mathcal{P} = (\sqrt{6}\hat{c} + 2)\mathcal{P}.$$

Similarly, for all $0 \leq k \leq j - 1$, we can show
$$\|\mathbf{y}_k\| \leq (\sqrt{6}\hat{c} + 2)\mathcal{P}.$$
By the update of NAG, we have $\|\mathbf{u}_{j-1} - \mathbf{y}_{j-1}\| = \zeta\|\mathbf{y}_{j-1} - \mathbf{y}_{j-2}\|$. Then
$$\|\mathbf{u}_{j-1}\| \leq \|\mathbf{u}_{j-1} - \mathbf{y}_{j-1}\| + \|\mathbf{y}_{j-1}\| = \zeta\|\mathbf{y}_{j-1} - \mathbf{y}_{j-2}\| + \|\mathbf{y}_{j-1}\|$$
$$\leq \|\mathbf{y}_{j-1} - \mathbf{y}_{j-2}\| + \|\mathbf{y}_{j-1}\| \leq \sqrt{6\eta\mathcal{F}} + (\sqrt{6}\hat{c} + 2)\mathcal{P} \leq (\sqrt{6}\hat{c} + 3)\mathcal{P}.$$
Similarly, for all $0 \leq k \leq j - 1$, we can show
$$\|\mathbf{u}_k\| \leq (\sqrt{6}\hat{c} + 3)\mathcal{P}.$$
Next we bound $\|\mathbf{y}_j\|$. Since $\|\mathbf{y}_j - \mathbf{y}_{j-1}\| = \|\mathbf{u}_{j-1} - \mathbf{u}_{j-2} + \eta(\nabla\hat{f}(\mathbf{u}_{j-2}) - \nabla\hat{f}(\mathbf{u}_{j-1}))\| \leq (1 + \eta L_1)\|\mathbf{u}_{j-1} - \mathbf{u}_{j-2}\|$, then
$$\|\mathbf{y}_j\| \leq (1 + \eta L_1)\|\mathbf{u}_{j-1} - \mathbf{u}_{j-2}\| + \|\mathbf{y}_{j-1}\| \leq 5/4\|\mathbf{u}_{j-1} - \mathbf{u}_{j-2}\| + \|\mathbf{y}_{j-1}\|$$
$$\leq 5/4\|\mathbf{u}_{j-1}\| + 5/4\|\mathbf{u}_{j-2}\| + \|\mathbf{y}_{j-1}\| \leq (7\sqrt{6}\hat{c}/2 + 19/2)\mathcal{P},$$
where the first inequality uses $\eta L_1 \leq \frac{1}{4}$. By using the relationship $\|\mathbf{u}_j - \mathbf{y}_j\| = \zeta\|\mathbf{y}_j - \mathbf{y}_{j-1}\|$ in NAG, we get
$$\|\mathbf{u}_j\| \leq \zeta\|\mathbf{y}_j - \mathbf{y}_{j-1}\| + \|\mathbf{y}_j\| \leq \|\mathbf{y}_j - \mathbf{y}_{j-1}\| + \|\mathbf{y}_j\| \leq \|\mathbf{y}_{j-1}\| + 2\|\mathbf{y}_j\| \leq (8\sqrt{6}\hat{c} + 21)\mathcal{P}.$$
Therefore, by choosing $\hat{c} \geq 15$, we have shown
$$\|\mathbf{y}_k\| \leq 10\hat{c}\mathcal{P} \text{ and } \|\mathbf{u}_k\| \leq 21\hat{c}\mathcal{P}, \ 0 \leq k \leq j. \tag{27}$$
Next, we will show that $\mathbf{y}_j$ is a NC, i.e., $\mathbf{y}_j^\top \nabla^2 f(\mathbf{x})\mathbf{y}_j \leq -\widetilde{\Omega}(\gamma^3)$. We know that the inequality (26) doesn't hold for all $0 \leq k < \tau$, then by the analysis of Lemma 7 (in particular inequality (36) due to that (26) does not hold any $k = 0, \ldots, \tau - 1$ ), we have
$$\hat{f}(\mathbf{y}_{k+1}) + \frac{1}{2\eta}\|\mathbf{y}_{k+1} - \mathbf{y}_k\|^2 \leq \hat{f}(\mathbf{y}_k) + \frac{1}{2\eta}\|\mathbf{y}_k - \mathbf{y}_{k-1}\|^2.$$
By summing up over $k$ from 0 to $j - 1$, we get
$$\hat{f}(\mathbf{y}_j) + \frac{1}{2\eta}\|\mathbf{y}_j - \mathbf{y}_{j-1}\|^2 \leq \hat{f}(\mathbf{y}_0).$$
Combining with $\|\mathbf{y}_j - \mathbf{y}_{j-1}\| \geq \sqrt{6\eta\mathcal{F}}$, we have
$$f(\mathbf{x} + \mathbf{y}_j) - f(\mathbf{x}) - \mathbf{y}_j^\top \nabla f(\mathbf{x}) \leq f(\mathbf{x} + \mathbf{y}_0) - f(\mathbf{x}) - \mathbf{y}_0^\top \nabla f(\mathbf{x}) - 3\mathcal{F}$$
$$\leq -3\mathcal{F} + \frac{L_1}{2}\|\mathbf{y}_0\|^2 \leq -3\mathcal{F} + \mathcal{F} \leq -2\mathcal{F},$$
By the Lipschitz continuity of Hessian, we have
$$\left| f(\mathbf{x} + \mathbf{y}_j) - f(\mathbf{x}) - \nabla f(\mathbf{x})^\top \mathbf{y}_j - \frac{1}{2}\mathbf{y}_j^\top \nabla^2 f(\mathbf{x})\mathbf{y}_j \right| \leq \frac{L_2}{6}\|\mathbf{y}_j\|^3.$$
Then
$$\frac{1}{2}\mathbf{y}_j^\top \nabla^2 f(\mathbf{x})\mathbf{y}_j \leq f(\mathbf{x} + \mathbf{y}_j) - f(\mathbf{x}) - \nabla f(\mathbf{x})^\top \mathbf{y}_j + \frac{L_2}{6}\|\mathbf{y}_j\|^3 \leq -2\mathcal{F} + \mathcal{F} \leq -\mathcal{F}.$$
Therefore, combining with (27) we get
$$\frac{\mathbf{y}_j^\top \nabla^2 f(\mathbf{x})\mathbf{y}_j}{\|\mathbf{y}_j\|^2} \leq -\frac{2\mathcal{F}}{(10\hat{c}\mathcal{P})^2} = -\frac{\gamma}{50\hat{c}^2 \log(dL_1/(\gamma\delta))} \leq -\frac{\gamma}{72\hat{c}^2 \log(dL_1/(\gamma\delta))}. \tag{28}$$

**Scenario (2)** For all $0 \leq k \leq \tau$, $\|\mathbf{y}_k - \mathbf{y}_{k-1}\| \leq \sqrt{6\eta\mathcal{F}}$. Thus,
$$\|\mathbf{y}_\tau\| \leq \sum_{k=1}^{\tau}\|\mathbf{y}_k - \mathbf{y}_{k-1}\| + \|\mathbf{y}_0\| \leq j\sqrt{6\eta\mathcal{F}} + 2\mathcal{P} \leq \hat{c}\mathcal{J}\sqrt{6\eta\mathcal{F}} + 2\mathcal{P} = (\sqrt{6}\hat{c} + 2)\mathcal{P} \leq 3\hat{c}\mathcal{P}.$$
Similarly, we can show
$$\|\mathbf{y}_k\| \leq 3\hat{c}\mathcal{P}, \ 0 \leq k \leq \tau, \tag{29}$$
where $\hat{c} \geq 15$. By the update of NAG, we have $\|\mathbf{u}_{j-1} - \mathbf{y}_{j-1}\| = \zeta\|\mathbf{y}_{j-1} - \mathbf{y}_{j-2}\|$. Then
$$\|\mathbf{u}_\tau\| \leq \|\mathbf{u}_\tau - \mathbf{y}_\tau\| + \|\mathbf{y}_\tau\| = \zeta\|\mathbf{y}_\tau - \mathbf{y}_{\tau-1}\| + \|\mathbf{y}_\tau\|$$
$$\leq \|\mathbf{y}_\tau - \mathbf{y}_{\tau-1}\| + \|\mathbf{y}_\tau\| \leq \sqrt{6\eta\mathcal{F}} + (\sqrt{6}\hat{c} + 2)\mathcal{P} \leq (\sqrt{6}\hat{c} + 3)\mathcal{P} \leq 3\hat{c}\mathcal{P}.$$

Similarly, we can show

$$\|\mathbf{u}_k\| \leq 3\hat{c}\mathcal{P},\ 0 \leq k \leq \tau. \tag{30}$$

In addition, we also have

$$\|\mathbf{y}_k - \mathbf{u}_k\| \leq 6\hat{c}\mathcal{P},\ 0 \leq k \leq \tau. \tag{31}$$

By expanding $\hat{f}(\mathbf{y}_\tau)$ in a Taylor series, we get

$$\hat{f}(\mathbf{y}_\tau) = \hat{f}(\mathbf{u}_\tau) + \nabla\hat{f}(\mathbf{u}_\tau)^\top(\mathbf{y}_\tau - \mathbf{u}_\tau) + \frac{1}{2}(\mathbf{y}_\tau - \mathbf{u}_\tau)^\top\nabla^2\hat{f}(\pi_\tau)(\mathbf{y}_\tau - \mathbf{u}_\tau),$$

where $\pi_\tau = \theta'\mathbf{u}_\tau + (1 - \theta')\mathbf{y}_\tau$ and $0 \leq \theta' \leq 1$. By inequality (26), we then have

$$\frac{1}{2}(\mathbf{y}_\tau - \mathbf{u}_\tau)^\top\nabla^2\hat{f}(\pi_\tau)(\mathbf{y}_\tau - \mathbf{u}_\tau) < -\frac{\gamma}{2}\|\mathbf{y}_\tau - \mathbf{u}_\tau\|^2.$$

It is clear that $\mathbf{y}_\tau - \mathbf{u}_\tau \neq 0$. Then we have

$$
\begin{aligned}
(\mathbf{y}_\tau - \mathbf{u}_\tau)^\top\nabla^2 f(\mathbf{x})(\mathbf{y}_\tau - \mathbf{u}_\tau) &\leq -\gamma\|\mathbf{y}_\tau - \mathbf{u}_\tau\|^2 + (\mathbf{y}_\tau - \mathbf{u}_\tau)^\top[\nabla^2 f(\mathbf{x}) - \nabla^2\hat{f}(\pi_\tau)](\mathbf{y}_\tau - \mathbf{u}_\tau) \\
&\leq -\gamma\|\mathbf{y}_\tau - \mathbf{u}_\tau\|^2 + L_2\|\theta'\mathbf{u}_\tau + (1 - \theta')\mathbf{y}_\tau\|\|\mathbf{y}_\tau - \mathbf{u}_\tau\|^2 \\
&\leq [-\gamma + L_2(\|\mathbf{u}_\tau\| + \|\mathbf{y}_\tau\|)]\|\mathbf{y}_\tau - \mathbf{u}_\tau\|^2 \\
&\leq -\frac{\gamma}{2}\|\mathbf{y}_\tau - \mathbf{u}_\tau\|^2.
\end{aligned}
$$

Therefore,

$$\frac{(\mathbf{y}_\tau - \mathbf{u}_\tau)^\top\nabla^2 f(x)(\mathbf{y}_\tau - \mathbf{u}_\tau)}{\|\mathbf{y}_\tau - \mathbf{u}_\tau\|^2} \leq -\frac{\gamma}{2} \leq -\frac{\gamma}{72\hat{c}^2\log(dL_1/(\gamma\delta))}. \tag{32}$$

Next, we will prove Theorem 2 under the condition that NCFind is not excuted, i.e., (26) does not happen.

**Lemma 7.** *Define* $\mathcal{F} = \eta L_1\frac{\gamma^3}{L_2^2} \cdot \log^{-3}(d\kappa/\delta)$, $\mathcal{P} = \sqrt{\eta L_1}\frac{\gamma}{L_2} \cdot \log^{-1}(d\kappa/\delta)$, $\mathcal{J} = \sqrt{\frac{\log(d\kappa/\delta)}{\eta\gamma}}$. *Suppose the inequality (26) doesn't hold. There exist a universal constant* $c_{\max}$, $\hat{c}$: *for* $\mathbf{x}$ *satisfies the condition that* $\lambda_{\min}(\nabla^2 f(\mathbf{x})) \leq -\gamma$, *any* $\|\mathbf{u}_0\| \leq 2r$, *where* $r = \frac{\mathcal{P}}{\kappa}\log^{-1}(d\kappa/\delta)$, *and*

$$T = \min\left\{\inf_\tau\{\tau|\hat{f}(\mathbf{y}_\tau) - \hat{f}(\mathbf{y}_0) \leq -3\mathcal{F}\}, \hat{c}\mathcal{J}\right\},$$

*If* $\zeta$ *satisfies* $1 - \frac{\sqrt{1+3\eta L_1}}{1-(1-s)\eta L_1}\zeta \geq \sqrt{\eta\gamma}$, *then for any* $\eta \leq c_{\max}/L_1$, *we have for all* $\tau < T$ *that* $\|\mathbf{u}_\tau\| \leq 6\hat{c}\mathcal{P}$.

**Lemma 8.** *There exist constant* $c_{\max}$, $\hat{c}$ *and a vector* $\mathbf{e}$ *with* $\|\mathbf{e}\| \leq 1$ *such that: for* $\mathbf{x}$ *satisfies the condition that* $\lambda_{\min}(\nabla^2 f(\mathbf{x})) \leq -\gamma$ *define a sequence* $\mathbf{w}_t$ *similar to* $\mathbf{u}_t$ *except* $\mathbf{w}_0 = \mathbf{u}_0 + \mu r\mathbf{e}$, *where* $\mu \in [\delta/2\sqrt{d}, 1]$, *and define a difference sequence* $\mathbf{v}_t = \mathbf{w}_t - \mathbf{u}_t$. *Let*

$$T = \min\{\inf_\tau\{\tau|\hat{f}_\mathbf{x}(\mathbf{y}_\tau) - \hat{f}_\mathbf{x}(\mathbf{y}_0) \leq -3\mathcal{F}\}, \hat{c}\mathcal{J}\}.$$

*where* $\mathbf{y}_\tau$ *is the sequence generated by (8) starting with* $\mathbf{w}_0$. *Then for any* $\eta \leq c_{\max}/L_1$, *if* $\|\mathbf{u}_\tau\| \leq 6\hat{c}\mathcal{P}$ *for all* $\tau < T$, *then* $T < \hat{c}\mathcal{J}$.

## I.1 Continuing the Proof of Theorem 2

*Proof.* When NCFind doesn't terminate the algorithm, then we use Lemma 7 and Lemma 8 to prove that NEON$^+$ finds NC with a high probability. Let us define $T_* = \hat{c}\mathcal{J}$ and $T' = \inf_\tau\{\tau|\hat{f}_\mathbf{x}(\mathbf{y}_\tau) - \hat{f}_\mathbf{x}(\mathbf{y}_0) \leq -3\mathcal{F}\}$, and consider following two scenarios:

**Scenario (1)** $T' \leq T_*$: By the unified update in (8), we have

$$\mathbf{y}_{\tau+1} = \mathbf{u}_\tau - \eta\nabla\hat{f}_\mathbf{x}(\mathbf{u}_\tau)$$

Then

$$\|\mathbf{y}_{T'}\| \leq (1 + \eta L_1)\|\mathbf{u}_{T'-1}\| \leq 12\hat{c}\mathcal{P} \triangleq U$$

By employing Lemma 7, we have

$$
\begin{aligned}
f(\mathbf{x} + \mathbf{y}_{T'}) - f(\mathbf{x}) - \mathbf{y}_{T'}^\top\nabla f(\mathbf{x}) &\leq f(\mathbf{x} + \mathbf{y}_0) - f(\mathbf{x}) - \mathbf{y}_0^\top\nabla f(\mathbf{x}) - 3\mathcal{F} \\
&\leq -3\mathcal{F} + \frac{L_1}{2}\|\mathbf{y}_0\|^2 \leq -3\mathcal{F} + \mathcal{F} \leq -2\mathcal{F}, \tag{33}
\end{aligned}
$$

Therefore we have

$$\min_{\substack{1 \leq \tau' \leq T_* \\ \|\mathbf{y}_{\tau'}\| \leq U, \mathbf{y}_\tau \sim \mathbf{u}_0}} f(\mathbf{x} + \mathbf{y}_{\tau'}) - f(\mathbf{x}) - \mathbf{y}_{\tau'}^\top \nabla f(\mathbf{x}) \leq f(\mathbf{x} + \mathbf{y}_{T'}) - f(\mathbf{x}) - \mathbf{y}_{T'}^\top \nabla f(\mathbf{x}) \leq -2\mathcal{F}.$$

where $\mathbf{y}_\tau \sim \mathbf{u}_0$ means that the generated sequence is starting from $\mathbf{u}_0$.

**Scenario (2)** $T' > T_*$**:** By Lemma 7, we have $\|\mathbf{u}_\tau\| \leq 6\mathcal{P}\hat{c}$ for all $\tau \leq T_*$. Let define $T'' = \inf_\tau \{\tau | \hat{f}_\mathbf{x}(\mathbf{y}_\tau) - \hat{f}_\mathbf{x}(\mathbf{y}_0) \leq -3\mathcal{F}\}$. Here, we abuse the same notation $\mathbf{y}_\tau$ to denote the generated sequence starting from $\mathbf{w}_0$. By Lemma 8, we konw $T'' \leq T_*$. From the proof of Lemma 8, we also know that $\|\mathbf{w}_{T''-1}\| \leq 6\mathcal{P}\hat{c}$. Hence $\|\mathbf{y}_{T''}\| \leq 12\hat{c}\mathcal{P}$. Similar to (33), we have

$$\min_{\substack{1 \leq \tau' \leq T_* \\ \|\mathbf{y}_{\tau'}\| \leq U, \mathbf{y}_\tau \sim \mathbf{w}_0}} f(\mathbf{x} + \mathbf{y}_{\tau'}) - f(\mathbf{x}) - \mathbf{y}_{\tau'}^\top \nabla f(\mathbf{x}) \leq -2\mathcal{F}.$$

Therefore,

$$\min \left\{ \min_{\substack{1 \leq \tau' \leq T_* \\ \|\mathbf{y}_{\tau'}\| \leq U, \mathbf{y}_\tau \sim \mathbf{u}_0}} f(\mathbf{x} + \mathbf{y}_{\tau'}) - f(\mathbf{x}) - \mathbf{y}_{\tau'}^\top \nabla f(\mathbf{x}), \right.$$
$$\left. \min_{\substack{1 \leq \tau' \leq T_* \\ \|\mathbf{y}_{\tau'}\| \leq U, \mathbf{y}_\tau \sim \mathbf{w}_0}} f(\mathbf{x} + \mathbf{y}_{\tau'}) - f(\mathbf{x}) - \mathbf{y}_{\tau'}^\top \nabla f(\mathbf{x}) \right\} \leq -2\mathcal{F}.$$

Please recall that $\mathbf{u}_0$ is random vector following an uniform distribution over $\mathbb{B}_0(r)$ with radius $r = \mathcal{P}/(\kappa \cdot \log(d\kappa/\delta))$. Let $\mathcal{X}_s \subset \mathbb{B}_0(r)$ be the set of bad initial points such that $\min_{1 \leq \tau' \leq T_*, \|\mathbf{y}'_\tau\| \leq U} f(\mathbf{x} + \mathbf{y}_{\tau'}) - f(\mathbf{x}) - \mathbf{y}_{\tau'}^\top \nabla f(\mathbf{x}) > -2\mathcal{F}$ when $\mathbf{u}_0 \in \mathcal{X}_s$; otherwise $\min_{1 \leq \tau' \leq T_*, \|\mathbf{y}'_\tau\| \leq U} f(\mathbf{x} + \mathbf{y}_{\tau'}) - f(\mathbf{x}) - \mathbf{y}_{\tau'}^\top \nabla f(\mathbf{x}) \leq -2\mathcal{F}$ when $\mathbf{u}_0 \in \mathbb{B}_0(r) - \mathcal{X}_s$.

From our analysis, for any $\mathbf{u}_0 \in \mathcal{X}_s$, we have $(\mathbf{u}_0 \pm \mu r \mathbf{e}) \notin \mathcal{X}_s$ where $\mu \in [\frac{\delta}{2\sqrt{d}}, 1]$. Denote by $I_{\mathcal{X}_s}(\cdot)$ the indicator function of being inside set $\mathcal{X}_s$. We set $u^{(1)}$ as the component along $\mathbf{e}$ direction and $\mathbf{u}^{(-1)}$ as the remaining $d - 1$ dimensional vector, then the vector $\mathbf{u} = (u^{(1)}, \mathbf{u}^{(-1)})$. The volume of $\mathcal{X}_s$'s can be upper bounded as:

$$\text{Vol}(\mathcal{X}_s) = \int_{\mathbb{B}_0^{(d)}(r)} d\mathbf{u} \cdot I_{\mathcal{X}_s}(\mathbf{u}) = \int_{\mathbb{B}_0^{(d-1)}(r)} d\mathbf{u}^{(-1)} \int_{-\sqrt{r^2 - \|\mathbf{u}^{(-1)}\|^2}}^{\sqrt{r^2 - \|\mathbf{u}^{(-1)}\|^2}} du^{(1)} \cdot I_{\mathcal{X}_s}(\mathbf{u})$$

$$\leq \int_{\mathbb{B}_0^{(d-1)}(r)} d\mathbf{u}^{(-1)} \cdot 2\mu r \leq \int_{\mathbb{B}_0^{(d-1)}(r)} d\mathbf{u}^{(-1)} \cdot 2\frac{\delta}{2\sqrt{d}}r = \text{Vol}(\mathbb{B}_0^{(d-1)}(r))\frac{\delta r}{\sqrt{d}}$$

Then,

$$\frac{\text{Vol}(\mathcal{X}_s)}{\text{Vol}(\mathbb{B}_0^{(d)}(r))} \leq \frac{\text{Vol}(\mathbb{B}_0^{(d-1)}(r))\frac{\delta r}{\sqrt{d}}}{\text{Vol}(\mathbb{B}_0^{(d)}(r))} = \frac{\delta}{\sqrt{\pi d}}\frac{\Gamma(\frac{d}{2} + 1)}{\Gamma(\frac{d}{2} + \frac{1}{2})} \leq \frac{\delta}{\sqrt{\pi d}} \cdot \sqrt{\frac{d}{2} + \frac{1}{2}} \leq \delta,$$

where the second inequality is due to $\frac{\Gamma(x+1)}{\Gamma(x+1/2)} < \sqrt{x + \frac{1}{2}}$ for all $x \geq 0$. Thus, we have $\mathbf{u}_0 \notin \mathcal{X}_s$ with at least probability $1 - \delta$. Therefore, with probability at least $1 - \delta$

$$\min_{\substack{1 \leq \tau' \leq T_* \\ \|\mathbf{y}_{\tau'}\| \leq U, \mathbf{y}_\tau \sim \mathbf{u}_0}} f(\mathbf{x} + \mathbf{y}_{\tau'}) - f(\mathbf{x}) - \mathbf{y}_{\tau'}^\top \nabla f(\mathbf{x}) \leq -2\mathcal{F}. \tag{34}$$

Let us complete the proof. By the Lipschitz continuity of Hessian, we have

$$\left| f(\mathbf{x} + \mathbf{y}_\tau) - f(\mathbf{x}) - \nabla f(\mathbf{x})^\top \mathbf{y}_\tau - \frac{1}{2}\mathbf{y}_\tau^\top \nabla^2 f(\mathbf{x})\mathbf{y}_\tau \right| \leq \frac{L_2}{6}\|\mathbf{y}_\tau\|^3.$$

Then by inequality (34), we get

$$\frac{1}{2}\mathbf{y}_\tau^\top \nabla^2 f(\mathbf{x})\mathbf{y}_\tau \leq f(\mathbf{x} + \mathbf{y}_\tau) - f(\mathbf{x}) - \nabla f(\mathbf{x})^\top \mathbf{y}_\tau + \frac{L_2}{6}\|\mathbf{y}_\tau\|^3 \leq -2\mathcal{F} + \mathcal{F} \leq -\mathcal{F}.$$

Therefore,

$$\frac{\mathbf{y}_\tau^\top \nabla^2 f(\mathbf{x})\mathbf{y}_\tau}{\|\mathbf{y}_\tau\|^2} \leq -\frac{2\mathcal{F}}{(12\hat{c}\mathcal{P})^2} = -\frac{\gamma}{72\hat{c}^2 \log(dL_1/(\gamma\delta))}, \tag{35}$$

If NEON$^+$returns 0 it is not difficult to prove that $\lambda_{\min}(\nabla^2 f(\mathbf{x})) \geq -\gamma$ holds with high probability $1 - O(\delta)$ for a sufficiently small $\delta$ by Bayes theorem. $\square$

## I.2 Proof of Lemma 7

*Proof.* By using the smoothness of $\hat{f}(\mathbf{u})$, we have

$$\hat{f}(\mathbf{y}_{\tau+1}) \leq \hat{f}(\mathbf{u}_\tau) + \nabla\hat{f}(\mathbf{u}_\tau)^\top (\mathbf{y}_{\tau+1} - \mathbf{u}_\tau) + \frac{L_1}{2}\|\mathbf{y}_{\tau+1} - \mathbf{u}_\tau\|^2$$

$$= \hat{f}(\mathbf{u}_\tau) - \eta\|\nabla\hat{f}(\mathbf{u}_\tau)\|^2 + \frac{1}{2}\eta^2 L_1\|\nabla\hat{f}(\mathbf{u}_\tau)\|^2,$$

where the last equality uses the update (8). Since the NCFind never happens during $t$ iterations, then the inequality (26) doesn't hold for any $\tau$, thus we have

$$\hat{f}(\mathbf{y}_{\tau+1}) + \nabla\hat{f}(\mathbf{u}_\tau)^\top(\mathbf{y}_\tau - \mathbf{u}_\tau) - \frac{\gamma}{2}\|\mathbf{y}_\tau - \mathbf{u}_\tau\|^2 \leq \hat{f}(\mathbf{y}_\tau) - \eta\|\nabla\hat{f}(\mathbf{u}_\tau)\|^2 + \frac{1}{2}\eta^2 L_1\|\nabla\hat{f}(\mathbf{u}_\tau)\|^2.$$

By the update (8), we have

$$\|\mathbf{y}_{\tau+1} - \mathbf{y}_\tau\|^2 = \|\mathbf{u}_\tau - \mathbf{y}_\tau - \eta\nabla\hat{f}(\mathbf{u}_\tau)\|^2$$

$$= \|\mathbf{u}_\tau - \mathbf{y}_\tau\|^2 - 2\eta\nabla\hat{f}(\mathbf{u}_\tau)^\top(\mathbf{u}_\tau - \mathbf{y}_\tau) + \eta^2\|\nabla\hat{f}(\mathbf{u}_\tau)\|^2$$

and $\|\mathbf{u}_\tau - \mathbf{y}_\tau\| = \zeta\|\mathbf{y}_\tau - \mathbf{y}_{\tau-1}\|$. Thus,

$$\hat{f}(\mathbf{y}_{\tau+1}) + \frac{1}{2\eta}\|\mathbf{y}_{\tau+1} - \mathbf{y}_\tau\|^2 \leq \hat{f}(\mathbf{y}_\tau) - \frac{\eta(1 - \eta L_1)}{2}\|\nabla\hat{f}(\mathbf{u}_\tau)\|^2 + \frac{1 + \eta\gamma}{2\eta}\|\mathbf{u}_\tau - \mathbf{y}_\tau\|^2$$

$$\leq \hat{f}(\mathbf{y}_\tau) + \frac{1 + \eta\gamma}{2\eta}\|\mathbf{u}_\tau - \mathbf{y}_\tau\|^2$$

$$= \hat{f}(\mathbf{y}_\tau) + \frac{(1 + \eta\gamma)\zeta^2}{2\eta}\|\mathbf{y}_\tau - \mathbf{y}_{\tau-1}\|^2$$

$$= \hat{f}(\mathbf{y}_\tau) + \frac{1}{2\eta}\|\mathbf{y}_\tau - \mathbf{y}_{\tau-1}\|^2 - \frac{1 - (1 + \eta\gamma)\zeta^2}{2\eta}\|\mathbf{y}_\tau - \mathbf{y}_{\tau-1}\|^2$$

$$\leq \hat{f}(\mathbf{y}_\tau) + \frac{1}{2\eta}\|\mathbf{y}_\tau - \mathbf{y}_{\tau-1}\|^2 - \frac{\sqrt{\eta\gamma}}{2\eta}\|\mathbf{y}_\tau - \mathbf{y}_{\tau-1}\|^2. \tag{36}$$

where the second inequality uses $\eta L_1 < 1$; and the last inequality uses $1 - (1 + \eta\gamma)\zeta^2 \geq 1 - \zeta$ by choosing $\zeta = 1 - \sqrt{\eta\gamma}$ with $\sqrt{\eta\gamma} \leq 1/2$.

By summing up $\tau$ from 0 to $t - 1$ where $t < T$, we have

$$\hat{f}(\mathbf{y}_t) + \frac{1}{2\eta}\|\mathbf{y}_t - \mathbf{y}_{t-1}\|^2 \leq \hat{f}(\mathbf{y}_0) - \frac{\sqrt{\eta\gamma}}{2\eta}\sum_{\tau=0}^{t-1}\|\mathbf{y}_\tau - \mathbf{y}_{\tau-1}\|^2,$$

which inplies

$$\frac{\sqrt{\eta\gamma}}{2\eta}\sum_{\tau=0}^{t-1}\|\mathbf{y}_\tau - \mathbf{y}_{\tau-1}\|^2 \leq \hat{f}(\mathbf{y}_0) - \hat{f}(\mathbf{y}_t).$$

Hence, if $t < T$ then $\hat{f}(\mathbf{y}_t) - \hat{f}(\mathbf{y}_0) \geq -3\mathcal{F}$, i.e., $\hat{f}(\mathbf{y}_0) - \hat{f}(\mathbf{y}_t) \leq 3\mathcal{F}$. Then we have

$$\sum_{\tau=0}^{t-1}\|\mathbf{y}_\tau - \mathbf{y}_{\tau-1}\| \leq \sqrt{t\sum_{\tau=0}^{t-1}\|\mathbf{y}_\tau - \mathbf{y}_{\tau-1}\|^2} \leq \sqrt{\frac{3\hat{c}\log^{1/2}(\frac{d\kappa}{\delta})}{\sqrt{\eta\gamma}}\frac{\eta L_1\gamma^3}{L_2^2\log^3(\frac{d\kappa}{\delta})}\frac{2\eta}{\sqrt{\eta\gamma}}}$$

$$\leq \sqrt{\frac{3\hat{c}}{\sqrt{\eta\gamma}}\frac{\eta L_1\gamma^3}{L_2^2\log^2(\frac{d\kappa}{\delta})}\frac{2\eta}{\sqrt{\eta\gamma}}} = \sqrt{6\hat{c}}\sqrt{\eta L_1}\frac{\gamma}{L_2}\log^{-1}(d\kappa/\delta) = \sqrt{6\hat{c}}\mathcal{P}.$$

Since $\sum_{\tau=1}^{t-1}\|\mathbf{u}_\tau - \mathbf{u}_{\tau-1}\| \leq \sum_{\tau=0}^{t-1}\|\mathbf{y}_{\tau+1} - \mathbf{y}_\tau\|$, then $\|\mathbf{u}_\tau\| \leq \sum_{t=1}^{\tau}\|\mathbf{u}_t - \mathbf{u}_{t-1}\| + \|\mathbf{u}_0\| \leq \sqrt{6\hat{c}}\mathcal{P} + \mathcal{P} \leq \hat{c}\mathcal{P}$, for all $\tau \leq t - 1$, where the last inequality is due to $\hat{c} \geq 43$. Additionally, by the unified update in (8), we have

$$\|\mathbf{u}_{\tau+1}\| \leq (1 + \eta L_1)\|\mathbf{u}_\tau\| + \zeta(1 + \eta L_1)\|\mathbf{u}_\tau\| + \zeta(1 + \eta L_1)\|\mathbf{u}_{\tau-1}\| \leq 6\hat{c}\mathcal{P}.$$

$\square$

## I.3   Proof of Lemma 8

*Proof.* The techniques for proving this lemma are largely borrowed from [12]. In particular, the proof is almost the same as the proof of Lemma 18 in [12]. For completeness, we include it in this subsection. First, the update of NAG can be written as

$$
\begin{aligned}
\mathbf{u}_{\tau+1} + \mathbf{v}_{\tau+1} &= \mathbf{w}_{\tau+1} \\
&= (1+\zeta)[\mathbf{w}_\tau - \eta(\nabla f(\mathbf{x}+\mathbf{w}_\tau) - \nabla f(\mathbf{x}))] - \zeta[\mathbf{w}_{\tau-1} - \eta(\nabla f(\mathbf{x}+\mathbf{w}_{\tau-1}) - \nabla f(\mathbf{x}))] \\
&= (1+\zeta)[\mathbf{u}_\tau + \mathbf{v}_\tau - \eta(\nabla f(\mathbf{x}+\mathbf{u}_\tau+\mathbf{v}_\tau) - \nabla f(\mathbf{x}))] \\
&\quad - \zeta[\mathbf{u}_{\tau-1} + \mathbf{v}_{\tau-1} - \eta(\nabla f(\mathbf{x}+\mathbf{u}_{\tau-1}+\mathbf{v}_{\tau-1}) - \nabla f(\mathbf{x}))] \\
&= (1+\zeta)[\mathbf{u}_\tau - \eta(\nabla f(\mathbf{x}+\mathbf{u}_\tau) - \nabla f(\mathbf{x}))] - \zeta[\mathbf{u}_{\tau-1} - \eta(\nabla f(\mathbf{x}+\mathbf{u}_{\tau-1}) - \nabla f(\mathbf{x}))] \\
&\quad + (1+\zeta)[\mathbf{v}_\tau - \eta H\mathbf{v}_\tau - \eta\Delta_\tau\mathbf{v}_\tau] - \zeta[\mathbf{v}_{\tau-1} - \eta H\mathbf{v}_{\tau-1} - \eta\Delta_{\tau-1}\mathbf{v}_{\tau-1}],
\end{aligned}
$$

where $\Delta_\tau = \int_0^1 \nabla^2 f(\mathbf{x}+\mathbf{u}_\tau+\theta\mathbf{v}_\tau))\mathrm{d}\theta - H$. Then

$$
\mathbf{v}_{\tau+1} = (1+\zeta)[\mathbf{v}_\tau - \eta H\mathbf{v}_\tau - \eta\Delta_\tau\mathbf{v}_\tau] - \zeta[\mathbf{v}_{\tau-1} - \eta H\mathbf{v}_{\tau-1} - \eta\Delta_{\tau-1}\mathbf{v}_{\tau-1}].
$$

and

$$
\begin{bmatrix} \mathbf{v}_{\tau+1} \\ \mathbf{v}_\tau \end{bmatrix} = \underbrace{\begin{bmatrix} (1+\zeta)(I-\eta H) & -\zeta(I-\eta H) \\ I & 0 \end{bmatrix}}_{A} \begin{bmatrix} \mathbf{v}_\tau \\ \mathbf{v}_{\tau-1} \end{bmatrix} - \eta \begin{bmatrix} (1+\zeta)\Delta_\tau\mathbf{v}_\tau - \zeta\Delta_{\tau-1}\mathbf{v}_{\tau-1} \\ 0 \end{bmatrix},
$$

where $\Delta_\tau = \int_0^1 \nabla^2 f(\mathbf{x}+\mathbf{u}_\tau+\theta\mathbf{v}_\tau))\mathrm{d}\theta - H$. It is easy to show that $\|\Delta_\tau\| \le L_2(\|\mathbf{u}_\tau\| + \|\mathbf{v}_\tau\|) \le 18L_2\mathcal{P}\hat{c}$. Let $\delta_\tau = (1+\zeta)\Delta_\tau\mathbf{v}_\tau - \zeta\Delta_{\tau-1}\mathbf{v}_{\tau-1}$, then

$$
\begin{bmatrix} \mathbf{v}_{\tau+1} \\ \mathbf{v}_\tau \end{bmatrix} = A \begin{bmatrix} \mathbf{v}_\tau \\ \mathbf{v}_{\tau-1} \end{bmatrix} - \eta \begin{bmatrix} \delta_\tau \\ 0 \end{bmatrix} = A^{\tau+1} \begin{bmatrix} \mathbf{v}_0 \\ \mathbf{v}_{-1} \end{bmatrix} - \eta \sum_{i=0}^\tau A^{\tau-i} \begin{bmatrix} \delta_i \\ 0 \end{bmatrix}. \tag{37}
$$

We can write $\mathbf{v}_\tau$ by (37) as follows :

$$
\mathbf{v}_\tau = [\ I \quad 0\ ] A^\tau \begin{bmatrix} \mathbf{v}_0 \\ \mathbf{v}_0 \end{bmatrix} - \eta [\ I \quad 0\ ] \sum_{i=0}^{\tau-1} A^{\tau-1-i} \begin{bmatrix} \delta_i \\ 0 \end{bmatrix},
$$

where uses the fact that $\mathbf{v}_{-1} = \mathbf{v}_0$. Next we will show the following inequality by induction:

$$
\frac{1}{2} \left\| [\ I \quad 0\ ] A^\tau \begin{bmatrix} \mathbf{v}_0 \\ \mathbf{v}_0 \end{bmatrix} \right\| \ge \left\| \eta [\ I \quad 0\ ] \sum_{i=0}^{\tau-1} A^{\tau-1-i} \begin{bmatrix} \delta_i \\ 0 \end{bmatrix} \right\| \tag{38}
$$

It is easy to show that the above inequality (38) holds for $\tau = 0$. We assume the inequality (38) holds for all $\tau$. Then,

$$
\begin{aligned}
\|\mathbf{v}_\tau\| &\le \left\| [\ I \quad 0\ ] A^\tau \begin{bmatrix} \mathbf{v}_0 \\ \mathbf{v}_0 \end{bmatrix} \right\| + \left\| \eta [\ I \quad 0\ ] \sum_{i=0}^{\tau-1} A^{\tau-1-i} \begin{bmatrix} \delta_i \\ 0 \end{bmatrix} \right\| \\
&\le \frac{3}{2} \left\| [\ I \quad 0\ ] A^\tau \begin{bmatrix} \mathbf{v}_0 \\ \mathbf{v}_0 \end{bmatrix} \right\|.
\end{aligned}
$$

On the other hand, we have $\|\delta_\tau\| \le (1+\zeta)\|\Delta_\tau\|\|\mathbf{v}_\tau\| + \zeta\|\Delta_{\tau-1}\|\|\mathbf{v}_{\tau-1}\| \le 54L_2\mathcal{P}\hat{c}(\|\mathbf{v}_\tau\| + \|\mathbf{v}_{\tau-1}\|)$, then we get

$$
\begin{aligned}
\|\delta_\tau\| &\le 54L_2\mathcal{P}\hat{c}(\|\mathbf{v}_\tau\| + \|\mathbf{v}_{\tau-1}\|) \\
&\le 81L_2\mathcal{P}\hat{c} \left( \left\| [\ I \quad 0\ ] A^\tau \begin{bmatrix} \mathbf{v}_0 \\ \mathbf{v}_0 \end{bmatrix} \right\| + \left\| [\ I \quad 0\ ] A^{\tau-1} \begin{bmatrix} \mathbf{v}_0 \\ \mathbf{v}_0 \end{bmatrix} \right\| \right) \\
&\le 162L_2\mathcal{P}\hat{c} \left\| [\ I \quad 0\ ] A^\tau \begin{bmatrix} \mathbf{v}_0 \\ \mathbf{v}_0 \end{bmatrix} \right\|,
\end{aligned}
$$

where the last inequality uses the monotonic property in terms of $\tau$ in Lemma 33 in [12] . We then need to prove the inequality (38) holds for $\tau + 1$. We consider the following term for $\tau + 1$:

$$\left\| \eta \begin{bmatrix} I & 0 \end{bmatrix} \sum_{i=0}^{\tau} A^{\tau-i} \begin{bmatrix} \delta_i \\ 0 \end{bmatrix} \right\| \leq \eta \sum_{i=0}^{\tau} \left\| \begin{bmatrix} I & 0 \end{bmatrix} A^{\tau-i} \begin{bmatrix} I \\ 0 \end{bmatrix} \right\| \|\delta_i\|$$

$$\leq 162\eta L_2 \mathcal{P}\hat{c} \sum_{i=0}^{\tau} \left\{ \left\| \begin{bmatrix} I & 0 \end{bmatrix} A^{\tau-i} \begin{bmatrix} I \\ 0 \end{bmatrix} \right\| \left\| \begin{bmatrix} I & 0 \end{bmatrix} A^i \begin{bmatrix} \mathbf{v}_0 \\ \mathbf{v}_0 \end{bmatrix} \right\| \right\}. \tag{39}$$

We know from the preconditions that $\lambda_{\min}(H) \leq -\gamma$ and the coordinate $\mathbf{e}_1$ is along the minimum eigenvector direction of Hessian matrix $H$, then we let the corresponding $2 \times 2$ matrix as $A_1$ and

$$\begin{bmatrix} a_\tau^{(1)} & -b_\tau^{(1)} \end{bmatrix} = \begin{bmatrix} 1 & 0 \end{bmatrix} A_1^\tau.$$

Since $\mathbf{v}_0 = \mathbf{w}_0 - \mathbf{u}_0 = \mu r \mathbf{e}_1$, $\mathbf{v}_0$ is along the $\mathbf{e}_1$ direction. By the analysis of Lemma 32 in [12], the matrix $\begin{bmatrix} I & 0 \end{bmatrix} A^{\tau-i} \begin{bmatrix} I \\ 0 \end{bmatrix}$ is diagonal, and thus the spectral norm is obtained along $\mathbf{e}_1$ which corresponding to $\lambda_{\min}(H)$. Thus, inequality (39) can be written as

$$\left\| \eta \begin{bmatrix} I & 0 \end{bmatrix} \sum_{i=0}^{\tau} A^{\tau-i} \begin{bmatrix} \delta_i \\ 0 \end{bmatrix} \right\| \leq \chi \sum_{i=0}^{\tau} \left\| \begin{bmatrix} I & 0 \end{bmatrix} A^{\tau-i} \begin{bmatrix} I \\ 0 \end{bmatrix} \right\| \left\| \begin{bmatrix} I & 0 \end{bmatrix} A^i \begin{bmatrix} \mathbf{v}_0 \\ \mathbf{v}_0 \end{bmatrix} \right\|$$

$$\leq \chi \sum_{i=0}^{\tau} a_{\tau-i}^{(1)}(a_i^{(1)} - b_i^{(1)})\|\mathbf{v}_0\| \leq \chi \sum_{i=0}^{\tau} \left( \frac{2}{1-\zeta} + (\tau+1) \right)(a_{\tau+1}^{(1)} - b_{\tau+1}^{(1)})\|\mathbf{v}_0\|$$

$$\leq \chi(\tau+1) \left( \frac{2}{1-\zeta} + (\tau+1) \right) \left\| \begin{bmatrix} I & 0 \end{bmatrix} A^{\tau+1} \begin{bmatrix} \mathbf{v}_0 \\ \mathbf{v}_0 \end{bmatrix} \right\|$$

$$\leq \chi(\hat{c}\mathcal{J}) \left( \frac{2}{1-\zeta} + \hat{c}\mathcal{J} \right) \left\| \begin{bmatrix} I & 0 \end{bmatrix} A^{\tau+1} \begin{bmatrix} \mathbf{v}_0 \\ \mathbf{v}_0 \end{bmatrix} \right\|,$$

where $\chi = 162\eta L_2 \mathcal{P}\hat{c}$. By choosing $2/(1-\zeta) \leq \hat{c}\mathcal{J}$, i.e. $\zeta \leq 1 - \frac{2\sqrt{\eta\gamma}}{\hat{c}\log(\kappa d/\delta)}$, then $162\eta L_2 \mathcal{P}\hat{c}(\hat{c}\mathcal{J})(2/(1-\zeta) + \hat{c}\mathcal{J}) \leq 312\eta L_2 \mathcal{P}\hat{c}(\hat{c}\mathcal{J})^2 = 312\hat{c}^3\sqrt{\eta L_1} = \frac{1}{2}$ if we select $\eta = \frac{c_{\max}}{L_1}$ with $c_{\max} = \frac{1}{624^2\hat{c}^6}$. Therefore, we have shown that the inequality (38) holds. Further, we have

$$\|\mathbf{v}_\tau\| \geq \left\| \begin{bmatrix} I & 0 \end{bmatrix} A^\tau \begin{bmatrix} \mathbf{v}_0 \\ \mathbf{v}_0 \end{bmatrix} \right\| - \left\| \eta \begin{bmatrix} I & 0 \end{bmatrix} \sum_{i=0}^{\tau-1} A^{\tau-1-i} \begin{bmatrix} \delta_i \\ 0 \end{bmatrix} \right\|$$

$$\geq \frac{1}{2} \left\| \begin{bmatrix} I & 0 \end{bmatrix} A^\tau \begin{bmatrix} \mathbf{v}_0 \\ \mathbf{v}_0 \end{bmatrix} \right\|$$

Noting that $\lambda_{\min}(H) \leq -\gamma$, by Lemmas 23, 30, 33 in [12], we have

$$\frac{1}{2} \left\| \begin{bmatrix} I & 0 \end{bmatrix} A^\tau \begin{bmatrix} \mathbf{v}_0 \\ \mathbf{v}_0 \end{bmatrix} \right\| \geq \begin{cases} \frac{\sqrt{|x|}}{4}(1 + \frac{\sqrt{|x|}}{2})^\tau \|\mathbf{v}_0\|, & x \in \left[ -\frac{1}{4}, -(1-\zeta)^2 \right], \\ \frac{1-\zeta}{4}(1 + \frac{|x|}{2(1-\zeta)})^\tau \|\mathbf{v}_0\|, & x \in \left[ -(1-\zeta)^2, 0 \right], \end{cases}$$

where $|x| = |\eta\lambda_{\min}(H)| \geq \eta\gamma$. By choosing $1 - \zeta = \sqrt{\eta\gamma}$, we have

$$\frac{1}{2} \left\| \begin{bmatrix} I & 0 \end{bmatrix} A_n^\tau \begin{bmatrix} \mathbf{v}_0 \\ \mathbf{v}_0 \end{bmatrix} \right\| \geq \frac{\sqrt{\eta\gamma}}{4} \left( 1 + \frac{\sqrt{\eta\gamma}}{2} \right)^\tau \mu r$$

Combining $\|\mathbf{v}_\tau\| \leq 12\mathcal{P}\hat{c}$, for all $\tau < T$, we have

$$12\mathcal{P}\hat{c} \geq \frac{\sqrt{\eta\gamma}}{4} \left( 1 + \frac{\sqrt{\eta\gamma}}{2} \right)^\tau \mu r = \frac{\sqrt{\eta\gamma}}{4} \left( 1 + \frac{\sqrt{\eta\gamma}}{2} \right)^\tau \mu \frac{\mathcal{P}}{\kappa} \log^{-1}(d\kappa/\delta)$$

$$\geq \frac{\sqrt{\eta\gamma}}{4} \left( 1 + \frac{\sqrt{\eta\gamma}}{2} \right)^\tau \frac{\delta}{2\sqrt{d}} \frac{\mathcal{P}}{\kappa} \log^{-1}(d\kappa/\delta)$$

$$= \frac{\sqrt{c_{\max}\gamma/L_1}}{4} \left( 1 + \frac{\sqrt{\eta\gamma}}{2} \right)^\tau \frac{\delta}{2\sqrt{d}} \frac{\mathcal{P}}{\kappa} \log^{-1}(d\kappa/\delta)$$

Then

$$T < \frac{\log(96\frac{\kappa^{3/2}\sqrt{d}}{\sqrt{c_{\max}}\delta} \cdot \hat{c}\log(d\kappa/\delta))}{\log(1 + \sqrt{\gamma\eta}/2)} \le \frac{\log(96\frac{\kappa^{3/2}\sqrt{d}}{\sqrt{c_{\max}}\delta} \cdot \hat{c}\log(d\kappa/\delta))}{2\sqrt{\gamma\eta}/3}$$
$$\le 1.5(2.5 + \log(96\hat{c}/\sqrt{c_{\max}}))\mathcal{J},$$

where the last inequality holds because of $\delta \in (0, \frac{d\kappa}{e}]$ and $\log(d\kappa/\delta) \ge 1$. By choosing $\hat{c} \ge 43$, we have $1.5(2.5 + \log(96\hat{c}/\sqrt{c_{\max}})) \le \hat{c}$, then $T < \hat{c}\mathcal{J}$ and complete the proof. $\qquad\square$

## J  Additional Simulation Results for Extracting NC

Figure 3: Comparison between different NEON procedures and Second-order Methods