[Reviews · NeurIPS 2018]

Reviewer 1



The paper proposes hessian-vector product free first-order stochastic algorithms that also have second-order convergence guarantee (converge to second order stationary points). The key advantage of the methods being that the complexity is linear in terms of the dimensionality of the problem. Related work has been discussed in detail and I particularly like the idea of putting Table1 at the very beginning to put the proposed method in perspective alongside other existing work. I also appreciated the fact that the authors acknowledged another very similar work that has appeared online. The authors have delved deeper into the theoretical guarantees and that is a plus of this work. I like the fact that Neon or Neon+ can be used as a black-box to replace the second-order methods for computing NC. This extends its applicability to variety of use-cases. The readability of the paper could be improved and it might help with some reorganization of content. It reads very verbosely to me at this point. The intuition and key results in certain theorems can be summarized in words to make it useful to the reader (for e.g. Theorem 2). Adding a Conclusion section with future directions of work would be helpful.

Reviewer 2



This paper proposes an efficient algorithm, Neon, to extract the negative-curvature of the Hessian matrix. Neon is very useful, which can be taken as a build block to convert existing convert existing stationary point finding algorithms into local minimum finding algorithms. This is because although the gradient of the points around saddle point is very small, there still exists a direction to decrease the function value by updating the variable along the negative-curvature of its Hessian matrix. In this way, Neon can be easily integrated with existing stationary point finding algorithms, such as SGD, accelerated SGD, SCGC, etc. Actually, the authors theoretically showed that SGD, SCGC, Natasha and SVRG can be combined with Neon and compared with existing works, such as noisy SGD and SGLD, the achieved computational complexity has much better dependence on the dimension d of the problem. Specifically, the complexity of noisy and SGLD rely on O(d^p) (p>=4), while the complexity in this work is only O(d). For the optimization accuracy, the best complexity in this work makes about improvement O(1/epsilon^0.75). Compared with Hessian vector product based algorithm, Neon only need to use first order information and is thus more practical. To my best knowledge, this work seems to be the first work which firstly use first-order gradient to compute the negative-curvature for escaping saddle points. Based on this work, there are some follow-up/independent works, such as Neon2 proposed by Allen-Zhu et al. So I think that this paper has very strong novelty and makes great contributions to optimization area, especially for escaping saddle points. However, the discussion of related works is not sufficient. For example, the differences between Neon and Neon2, AGD proposed by Jin Chi are not very clear, since they share similar techniques and the authors only slightly mentioned the differences. The authors claim that their work is the first one which only linearly depends on the problem dimension d. I wonder whether existing works, such as Natasha2, SNCG although they involve the second information, e.g. Hessian vector product, also depend on the problem dimension d polynomially or higher. Finally, as for the experiments, the authors only show the convergence behavior of Neon-SGD, Neon+-SGD and noisy SGD in Fig. 2. However, these plots only demonstrated that these algorithms can converge but cannot prove they can escape saddle points. A better comparison may be that adding an algorithm, e.g. SGD, and looking at SGD can achieve the same optimization accuracy with Neon-SGD, Neon+-SGD and noisy SGD.

Reviewer 3



This paper proposes a new algorithm for finding second order stationary points for stochastic, second-order smooth and nonconvex problems, using only first order informations. One of the key building blocks of this paper is the so-called NEON method to escape the strict saddle point. Overall, the big picture of this paper is clearly presented. However, many details and technical proofs are badly written, making overall the paper hard to evaluate. Below are some problems that I am interested in. (1) The authors of this paper are pretty careful in analyzing the dependence of their method with respect to the dimensionality $d$. However, I do think that the constants G and V in the Assumptions 1 (iii) and 1 (iv) are linearly dependent on $d$ in a bunch of applications. Please elaborate more onto this point in the rebuttal. (2) This paper seems to suggest that they are considering the problem under stochastic setting. However, the main contribution of this paper, to me, seems to lie within analyzing how to quickly escape saddle points fast under \emph{deterministic} setting. It seems to me the analysis here in Theorem 1 and Theorem 2 are new (while understanding that a large part of the proofs are adapted from Nesterov's acceleration paper and reference [11] and [12]), and the analysis here in Theorem 3 is basically a trivial combination of existing analysis for deterministic setting and some concentration bounds. Hence, I am a little bit confused about what the actual contribution of this paper is when reading this paper. (3) I really hope the author clarify more clearly about their contribution of this paper. Moreover, the writing of this paper needs to be strengthened. It took me a long time to figure out that the constants \mathcal{F, P, J, U, r} etc. on the negative curvature $\eta$ is changed from Theorem 1 to Theorem 2. Overall, I give a weak acceptance for this paper.

Reviewer 4



This paper considers first-order (FO) stochastic non-convex optimization problems. Authors propose FO procedures to extract negative curvature directions from the Hessian which are referred to as NEON. Next, they combine these with FO stochastic algorithms to find critical points. Time complexities, and dependence on the problem dimension is established for approximate second-order stationary points. The results of this paper are good, but the paper is not written carefully. I find the statements of the theorems not clear, and several important remarks are hard to parse. The paper requires significant additional work. Assumption 1-ii: Isn't this always true for any reasonable function? Assumption 1-iii: Please explicitly state the sub-gaussianity. I think it is OK to assume that the constant G does not depend on the dimension d due to rotation invariance of sub-gaussian random variables; however, I am not sure about the constant V in Assmp. 1-iv. In general, V (the variance) should be of order d as it is the trace of dxd matrix. I would be happy to see a discussion about these constants and their dependence on the dimension. When is this assumption used? How does the dependence of V to dimension d affect the claimed linear dependence result? Which constants are affected? Can you provide an argument why |\hat{x}_\tau| is expected to be small? Steps leading up to that point is clear, and this would improve readability. In Thm. 1, does c_max only depend on \hat{c}? NEON requires the knowledge of \mathcal{F}, hence the parametes L_1, L_2, \gamma, \delta. Please include a discussion on this: how would you estimate those in practice? Thm. 1,2,&3 operates under Asmp.1 (at least partially?), but this is not stated explicitly. The statements of the Thm.'s and the underlying assumptions are not transparent. For example in Thm.3, the reader cannot figure out the required mini-batch size to make the constants sufficiently small, whether these assumptions are reasonable or not. This is a very important conclusion of that theorem as it operates under mini batches. Line 170 would be a good place to explain why you are trying to match the complexity of Lanczos. A brief discussion after each thm would improve the presentation significantly. Statement in line 220 is very important but hard to parse. It would be great to see a detailed comparison with [2] in the updated version. - line 48: extract - line 87: Is [t] ever used? - line 88: sphere of an Euclidean ball, also perhaps use S_r^{d-1} - line 117: maximum eigen-vector - line 175: smaller than - ..